# LESS IS MORE: ONE-SHOT-SUBGRAPH LINK PREDICTION ON LARGE-SCALE KNOWLEDGE GRAPHS

**Zhanke Zhou**[1]   **Yongqi Zhang**[2]   **Jiangchao Yao**[3]   **Quanming Yao**[4]   **Bo Han**[1][†]
[1]TMLR Group, Hong Kong Baptist University
[2]The Hong Kong University of Science and Technology (Guangzhou)
[3]CMIC, Shanghai Jiao Tong University    [4]Tsinghua University
{cszkzhou, bhanml}@comp.hkbu.edu.hk    yzhangee@connect.ust.hk
sunarker@sjtu.edu.cn    qyaoaa@tsinghua.edu.cn

## ABSTRACT

To deduce new facts on a knowledge graph (KG), a link predictor learns from the graph structure and collects local evidence to find the answer to a given query. However, existing methods suffer from a severe scalability problem due to the utilization of the whole KG for prediction, which hinders their promise on large-scale KGs and cannot be directly addressed by vanilla sampling methods. In this work, we propose the *one-shot-subgraph* link prediction to achieve efficient and adaptive prediction. The design principle is that, instead of directly acting on the whole KG, the prediction procedure is decoupled into two steps, *i.e.*, (i) extracting *only one* subgraph according to the query and (ii) predicting on this single, query-dependent subgraph. We reveal that the non-parametric and computation-efficient heuristics Personalized PageRank (PPR) can effectively identify the potential answers and supporting evidence. With efficient subgraph-based prediction, we further introduce the automated searching of the optimal configurations in both data and model spaces. Empirically, we achieve promoted efficiency and leading performances on five large-scale benchmarks. The code is publicly available at: https://github.com/tmlr-group/one-shot-subgraph.

## 1 INTRODUCTION

A knowledge graph (KG) is graph-structural data with relational facts (Battaglia et al., 2018; Ji et al., 2020; Chen et al., 2020), based on which, one can conduct link prediction to deduce new facts from existing ones. The typical problem is to find the answer entity for the specific query, *e.g.*, to find the answer *Los Angeles* to the query *(LeBron, lives_in, ?)*. With continuous advances in recent years, the link prediction on KG has been widely applied in recommendation systems (Cao et al., 2019; Wang et al., 2019), online question answering (Huang et al., 2019), and drug discovery (Yu et al., 2021).

The prediction system learns from the local structure of a KG, where existing methods can be generally summarized as two categories: (1) *semantic* models that implicitly capture the local evidence through learning the low-dimensional embeddings of entities and relations (Bordes et al., 2013; Dettmers et al., 2017; Zhang et al., 2019; Wang et al., 2017); and (2) *structural* models that explicitly explore the KG's structure based on relational paths or graphs with recurrent neural networks (RNNs) or graph neural networks (GNNs) (Das et al., 2017; Schlichtkrull et al., 2018; Sadeghian et al., 2019; Vashishth et al., 2019; Teru et al., 2020; Qu et al., 2021; Zhu et al., 2021; Zhang & Yao, 2022).

Although achieving leading performances, these structural models suffer from a severe scalability problem as the entire KG has been potentially or progressively taken for prediction. This inefficient manner hinders their application and optimization on large-scale KGs, *e.g.*, OGB (Hu et al., 2020). Thus, it raises an open question: *Is all the information necessary for prediction on knowledge graphs?* Intuitively, only partial knowledge stored in the human brain is relevant to a given question, which is extracted by recalling and then utilized in the careful thinking procedure. Similarly, generating candidates and then ranking promising ones are common practices in large-scale recommendation systems with millions even billions of users (Cheng et al., 2016; Covington et al., 2016). These facts motivate us to conduct efficient link prediction with an effective sampling mechanism for KGs.

---

[†]Correspondence to Bo Han (bhanml@comp.hkbu.edu.hk).

In this work, we propose the novel *one-shot-subgraph* link prediction on a knowledge graph. This idea paves a new way to alleviate the scalability problem of existing KG methods from a data-centric perspective: decoupling the prediction procedure into two steps with a corresponding sampler and a predictor. Thereby, the prediction of a specific query is conducted by (i) *fast* sampling of one query-dependent subgraph with the sampler and (ii) *slow* prediction on the subgraph with predictor.

Nevertheless, it is non-trivial to achieve efficient and effective link prediction on large-scale KGs due to the two major challenges. (1) *Sampling speed and quality*: The fast sampling of the one-shot sampler should be capable of covering the essential evidence and potential answers to support the query. (2) *Joint optimization*: The sampler and predictor should be optimized jointly to avoid trivial solutions and to guarantee the expressiveness and adaptivity of the overall model to a specific KG.

To solve these challenges technically, we first implement the one-shot-subgraph link prediction by the non-parametric and computation-efficient Personalized PageRank (PPR), which is capable of effectively identifying the potential answers without requiring learning. With the efficient subgraph-based prediction, we further propose to search the data-adaptive configurations in both data and model spaces. We show it unnecessary to utilize the whole KG in inference; meanwhile, only a relatively small proportion of information (*e.g.*, 10% of entities) is sufficient. Our main contributions are:

- We conceptually formalize the new manner of *one-shot-subgraph link prediction* on KGs (Sec. 3) and technically instantiate it with efficient heuristic samplers and powerful KG predictors (Sec. 4.1).
- We solve a non-trivial and bi-level optimization problem of searching the optimal configurations in both data and model spaces (Sec. 4.2) and theoretically analyze the extrapolation power (Sec. 4.3).
- We conduct extensive experiments on five large-scale datasets and achieve an average of $94.4\%$ improvement in efficiency of prediction and $6.9\%$ promotion in effectiveness of prediction (Sec. 5).

## 2 PRELIMINARIES

**Notations.** A knowledge graph is denoted as $\mathcal{G} = (\mathcal{V}, \mathcal{R}, \mathcal{E})$, where the entity set $\mathcal{V}$, the relation set $\mathcal{R}$, and factual edge set $\mathcal{E} = \{(x, r, v) : x, v \in \mathcal{V}, r \in \mathcal{R}\}$. Here, a fact is formed as a triplet and denoted as $(x, r, v)$. Besides, a sampled *subgraph* of $\mathcal{G}$ is denoted as $\mathcal{G}_s = (\mathcal{V}_s, \mathcal{R}_s, \mathcal{E}_s)$, satisfying that $\mathcal{V}_s \subseteq \mathcal{V}, \mathcal{R}_s \subseteq \mathcal{R}, \mathcal{E}_s \subseteq \mathcal{E}$. The atomic problem of link prediction is denoted as a query $(u, q, ?)$, *i.e.*, given the query entity $u$ and query relation $q$, to find the answer entity $v$, making $(u, q, v)$ valid.

**Semantic models** encode entities and relations to low-dimensional entity embeddings $\boldsymbol{H}_{\mathcal{V}} \in \mathbb{R}^{|\mathcal{V}| \times D_v}$ and relation embeddings $\boldsymbol{H}_{\mathcal{R}} \in \mathbb{R}^{|\mathcal{R}| \times D_r}$, where $D_v, D_r$ are dimensions. A scoring function $f_{\boldsymbol{\theta}}$, *e.g.*, TransE (Bordes et al., 2013) or QuatE (Zhang et al., 2019), is necessary here to quantify the plausibility of a query triplet $(u, q, v)$ with the learned embeddings $(\boldsymbol{h}_u, \boldsymbol{h}_q, \boldsymbol{h}_v)$ as $f_{\boldsymbol{\theta}} : (\mathbb{R}^{D_v}, \mathbb{R}^{D_r}, \mathbb{R}^{D_v}) \mapsto \mathbb{R}$.

**Efficient semantic models** aim to reduce the size of entity embeddings. NodePiece (Galkin et al., 2022) proposes an anchor-based approach that obtains fixed-size embeddings as $\mathcal{G} \xmapsto{f_{\boldsymbol{\theta}}} \hat{\boldsymbol{H}}_{\mathcal{V}} \in \mathbb{R}^{N \times D_v}$ and inference as $(\hat{\boldsymbol{H}}, \mathcal{G}) \xmapsto{f_{\boldsymbol{\theta}}, (u,q)} \hat{\boldsymbol{Y}}$, where $\hat{\boldsymbol{Y}}$ are the scores of candidate answers, and $N \ll |\mathcal{V}|$. Designed to reduce the embedding size, NodePiece cannot reduce the graph size for structural models.

**Structural models** are based on relational *paths* or *graphs* for link prediction. Wherein the *path*-based models, *e.g.*, MINERVA (Das et al., 2017), DRUM (Sadeghian et al., 2019), and RNNLogic (Qu et al., 2021), aim to learn probabilistic and logical rules and well capture the sequential patterns in KGs. The *graph*-based models such as R-GCN (Schlichtkrull et al., 2018) and CompGCN (Vashishth et al., 2019) propagate the low-level entity embeddings among the neighboring entities to obtain high-level embeddings. Recent methods NBFNet (Zhu et al., 2021) and RED-GNN (Zhang & Yao, 2022) progressively propagate from $u$ to its neighborhood in a breadth-first-searching (BFS) manner.

**Sampling-based structural models** adopt graph sampling approaches to decrease the computation complexity, which can be categorized into two-fold as follows. First, *subgraph-wise* methods such as GraIL (Teru et al., 2020) and CoMPILE (Mai et al., 2021) extract enclosing subgraphs between query entity $u$ and each candidate answer $v$. Second, *layer-wise* sampling methods extract a subgraph for message propagation in each layer of a model. Wherein designed for node-level tasks on homogeneous graphs, GraphSAGE (Hamilton et al., 2017) and FastGCN (Chen et al., 2018) randomly sample neighbors around the query entity. While the KG sampling methods, *e.g.*, DPMPN (Xu et al., 2019), AdaProp (Zhang et al., 2023c), and AStarNet (Zhu et al., 2023), extract a learnable subgraph in $\ell$-th layer by the GNN model in $\ell$-th layer, coupling the procedures of sampling and prediction.

## 3 *One-shot-subgraph* LINK PREDICTION ON KNOWLEDGE GRAPHS

To achieve efficient link prediction, we conceptually design the *one-shot-subgraph* manner that avoids directly predicting with the entire KG. We formalize this new manner in the following Def. 1.

**Definition 1** (One-shot-subgraph Link Prediction on Knowledge Graphs). *Instead of directly predicting on the original graph $\mathcal{G}$, the prediction procedure is decoupled to two-fold: (1) one-shot sampling of a query-dependent subgraph and (2) prediction on this subgraph. The prediction pipeline becomes*

$$\mathcal{G} \xmapsto{g_\phi,(u,q)} \mathcal{G}_s \xmapsto{f_\theta} \hat{Y}, \tag{1}$$

*where the sampler $g_\phi$ generates only one subgraph $\mathcal{G}_s$ (satisfies $|\mathcal{V}_s| \ll |\mathcal{V}|, |\mathcal{E}_s| \ll |\mathcal{E}|$) conditioned on the given query $(u, q, ?)$. Based on subgraph $\mathcal{G}_s$, the predictor $f_\theta$ outputs the final predictions $\hat{Y}$.*

**Comparison with existing manners of prediction.** In brief, semantic models follow the manner of encoding the entire $\mathcal{G}$ to the embeddings $\boldsymbol{H} = (\boldsymbol{H}_\mathcal{V}, \boldsymbol{H}_\mathcal{R})$ and prediction (inference) without $\mathcal{G}$, *i.e.*,

$$\boldsymbol{H} \xmapsto{f_\theta,(u,q)} \hat{Y}, \text{ s.t. } \mathcal{G} \xmapsto{f_\theta} \boldsymbol{H},$$

which is parameter-expensive, especially when encountering a large-scale graph with a large entity set. On the other hand, structural models adopt the way of learning and prediction with $\mathcal{G}$, *i.e.*,

$$\mathcal{G} \xmapsto{f_\theta,(u,q)} \hat{Y},$$

that directly or progressively conduct prediction with the entire graph $\mathcal{G}$. Namely, all the entities and edges can be potentially taken in the prediction of one query, which is computation-expensive. By contrast, our proposed one-shot prediction manner (Def. 1) enjoys the advantages 1 & 2 as follows.

**Advantage 1** (Low complexity of computation and parameter). *The one-shot-subgraph model is (1) computation-efficient: the extracted subgraph is much smaller than the original graph, i.e., $|\mathcal{V}_s| \ll |\mathcal{V}|$ and $|\mathcal{E}_s| \ll |\mathcal{E}|$; and (2) parameter-efficient: it avoids learning the expensive entities' embeddings.*

**Advantage 2** (Flexible propagation scope). *The scope here refers to the range of message propagation starting from the query entity $u$. Normally, an L-layer structural method will propagate to the full L-hop neighbors of $u$. By contrast, the adopted one-shot sampling enables the bound of propagation scope within the extracted subgraph, where the scope is decoupled from the model's depth $L$.*

**Comparison with existing sampling methods.** Although promising to the scalability issue, existing sampling methods for structural models are not efficient or effective enough for learning and prediction on large-scale KGs. To be specific, the *random* layer-wise sampling methods cannot guarantee the coverage of answer entities, *i.e.*, $\mathbb{1}(v \in \mathcal{V}_u)$. By contrast, the *learnable* layer-wise sampling methods extract the query-dependent subgraph $\mathcal{G}_s^{(\ell)}$ in $\ell$-th layer via the GNN model $f_\theta^{(\ell)}$ in $\ell$-th layer as

$$\mathcal{G} \xmapsto{f_\theta^{(1)},(u,q)} \mathcal{G}_s^{(1)} \xmapsto{f_\theta^{(2)},(u,q)} \mathcal{G}_s^{(2)} \mapsto \cdots \mapsto \mathcal{G}_s^{(L-1)} \xmapsto{f_\theta^{(L)},(u,q)} \hat{Y},$$

coupling the sampling and prediction procedures that (1) are bundled with specific architectures and (2) with extra computation cost in the layer-wise sampling operation. Besides, the subgraph-wise sampling methods extract the enclosing subgraphs between query entity $u$ and each candidate answer $v \in \mathcal{V}$, and then independently reason on each of these subgraphs to obtain the final prediction $\hat{Y}$ as

$$\left\{ \hat{Y}_v : \mathcal{G} \xmapsto{(u,v)} \mathcal{G}_s^{(u,v)} \xmapsto{f_\theta,(u,q,v)} \hat{Y}_v \right\}_{v \in \mathcal{V}} \mapsto \hat{Y}.$$

Note these approaches are extremely expensive on large-scale graphs, as each candidate $(u, v)$ corresponds to a subgraph to be scored. By contrast, one-shot sampling manner enjoys the advantage 3.

**Advantage 3** (High efficiency in subgraph sampling). *Our proposed prediction manner requires **only one subgraph** for answering **one query**, which is expected to cover all the potential answers and supporting facts. Notably, this query-specific subgraph is extracted in a **one-shot** and **decoupled** manner that does not involve the predictor, reducing the computation cost in subgraph sampling.*

## 4 INSTANTIATING THE ONE-SHOT-SUBGRAPH LINK PREDICTION

Note that it is non-trivial to achieve Def. 1, wherein (i) the implementation of sampler, (ii) the architecture of predictor, and (iii) the method to optimize these two modules need to be figured out. Here, the major challenge lies in the sampler, which is required to be efficient, query-dependent, and local-structure-preserving. In this section, we elaborate on the detailed implementation (Sec. 4.1), set up a bi-level problem for optimization (Sec. 4.2), and investigate the extrapolation power (Sec. 4.3).

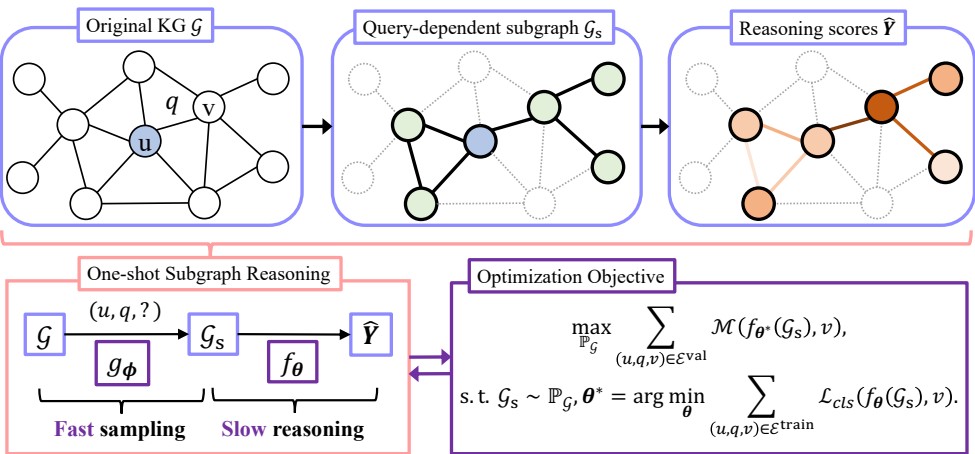

Figure 1: The proposed framework of one-shot-subgraph link prediction. Specifically, (1) the sampler $g_\phi$ extracts a subgraph $\mathcal{G}_s$ from the whole graph $\mathcal{G}$ with regard to the given query, and (2) the predictor $f_\theta$ conducts deliberative prediction on the extracted subgraph $\mathcal{G}_s$ and obtains the final predictions $\hat{Y}$.

## 4.1 REALIZATION: THREE-STEP PREDICTION WITH PERSONALIZED PAGERANK

**Overview.** As illustrated in Fig. 1, the three key steps of our method are **(1)** generating the sampling distribution $\mathbb{P}_\mathcal{G}$ by sampler $g_\phi$, **(2)** sampling a subgraph from the distribution as $\mathcal{G}_s \sim \mathbb{P}_\mathcal{G}$ with top-$K$ entities and edges, and **(3)** predicting on the subgraph $\mathcal{G}_s$ and acquiring the final prediction $\hat{Y}$ by predictor $f_\theta$. The three-step procedure is summarized in Algorithm 1 and elaborated on as follows.

**Step-1. Generate sampling distribution.** Previous studies show that $v$ is generally near to $u$ (Zhu et al., 2021; Zhang & Yao, 2022), and the relational paths connecting $u$ and $v$ that support the query also lie close to $u$ (Das et al., 2017; Sadeghian et al., 2019). To efficiently capture the local evidence of $u$, we choose the heuristic Personalized PageRank (PPR) (Page et al., 1999; Jeh & Widom, 2003) as the sampling indicator. Note that PPR is not only *efficient* for its non-parametric nature but also *query-dependent* and *local-structure-preserving* for its single-source scoring that starts from $u$.

Specifically, PPR starts propagation from $u$ to evaluate the importance of each neighbor of $u$ and generates the PageRank scores as the sampling probability that encodes the local neighborhood of the query entity $u$. Besides, it can also preserve the locality and connectivity of subgraphs by leveraging the information from a large neighborhood. Given a query entity $u$, we obtain the probability $\boldsymbol{p} \in \mathbb{R}^{|\mathcal{V}|}$

$$\texttt{Non-parametric indicator}: \boldsymbol{p}^{(k+1)} \leftarrow \alpha \cdot \boldsymbol{s} + (1-\alpha) \cdot \boldsymbol{D}^{-1} \boldsymbol{A} \cdot \boldsymbol{p}^{(k)}, \quad (2)$$

by iteratively updating the scores up to $K = 100$ steps to approximate the converged scores efficiently. Here, the initial score $\boldsymbol{p}^{(0)} = \boldsymbol{s} = \mathbb{1}(u) \in \{0,1\}^{|\mathcal{V}|}$ indicates the query entity $u$ to be explored. The two-dimensional degree matrix $\boldsymbol{D} \in \mathbb{R}^{|\mathcal{V}| \times |\mathcal{V}|}$ and adjacency matrix $\boldsymbol{A} \in \{0,1\}^{|\mathcal{V}| \times |\mathcal{V}|}$ together work as the transition matrix, wherein $\boldsymbol{A}_{ij} = 1$ means an edge $(i,r,j) \in \mathcal{E}$ and $\boldsymbol{D}_{ij} = \text{degree}(v_i)$ if $i = j$ else $\boldsymbol{D}_{ij} = 0$. The damping coefficient $\alpha$ ($= 0.85$ by default) controls the differentiation degree.

**Step-2. Extract a subgraph.** Based on the PPR scores $\boldsymbol{p}$ (Eqn. 2), the subgraph $\mathcal{G}_s = (\mathcal{V}_s, \mathcal{E}_s, \mathcal{R}_s)$ (where $\mathcal{R}_s = \mathcal{R}$) is extracted with the most important entities and edges. Denoting the sampling ratios of entities and edges as $r_\mathcal{V}^q, r_\mathcal{E}^q \in (0,1]$ that depend on the query relation $q$, we sample $|\mathcal{V}_s| = r_\mathcal{V}^q \times |\mathcal{V}|$ entities and $|\mathcal{E}_s| = r_\mathcal{E}^q \times |\mathcal{E}|$ edges from the full graph $\mathcal{G}$. With the $\texttt{TopK}(D, P, K)$ operation that picks up top-$K$ elements from candidate $D$ *w.r.t.* probability $P$, the entities $\mathcal{V}_s$ and edges $\mathcal{E}_s$ are given as

$$\begin{aligned} \texttt{Entity Sampling}:~ &\mathcal{V}_s \leftarrow \texttt{TopK}\Big(\mathcal{V},~ \boldsymbol{p},~ K = r_\mathcal{V}^q \times |\mathcal{V}|\Big), \\ \texttt{Edge Sampling}:~ &\mathcal{E}_s \leftarrow \texttt{TopK}\Big(\mathcal{E},~ \{\boldsymbol{p}_x \cdot \boldsymbol{p}_o : x, o \in \mathcal{V}_s, (x,r,o) \in \mathcal{E}\},~ K = r_\mathcal{E}^q \times |\mathcal{E}|\Big). \end{aligned} \quad (3)$$

**Step-3. Reason on the subgraph.** From the model's perspective, we build the configuration space of the predictor and further utilize the advantages of existing structural models introduced in Sec. 2. Three query-dependent message functions $\texttt{MESS}(\cdot)$ are considered, including DRUM, NBFNet, and RED-GNN, which are elaborated in Appendix. B. Note the effective message is propagated from $u$ to the sampled entities $o \in \mathcal{V}_s$. Generally, the layer-wise updating of representations is formulated as

$$\texttt{Indicating:} \boldsymbol{h}_o^{(0)} \leftarrow \mathbb{1}(o = u),$$

$$\texttt{Propagation:} \boldsymbol{h}_o^{(\ell+1)} \leftarrow \texttt{DROPOUT}\Big(\texttt{ACT}\Big(\texttt{AGG}\big\{\texttt{MESS}(\boldsymbol{h}_x^{(\ell)}, \boldsymbol{h}_r^{(\ell)}, \boldsymbol{h}_o^{(\ell)}) : (x, r, o) \in \mathcal{E}_s\big\}\Big)\Big), \quad (4)$$

where $\mathbb{1}(o = u)$ is the indicator function that only labels the query entity $u$ in a query-dependent manner. After a $L$-layer propagation, the predictor outputs the final score $\hat{\boldsymbol{y}}_o$ of each entity $o \in \mathcal{V}_s$ based on their representations $\boldsymbol{h}_o^{(L)}$ as $\hat{\boldsymbol{y}}_o = \texttt{Readout}(\boldsymbol{h}_o^{(L)}, \boldsymbol{h}_u^{(L)}) \in \mathbb{R}$. The loss function $\mathcal{L}_{cls}$ adopted in the training phase is the commonly-used binary cross-entropy loss on all the sampled entities. Namely, $\mathcal{L}_{cls} = -\sum_{o \in \mathcal{V}_s} y_o \log(\hat{\boldsymbol{y}}_o) + (1 - y_o) \log(1 - \hat{\boldsymbol{y}}_o)$, where $y_o = 1$ if $o = v$ else $y_o = 0$.

---

**Algorithm 1** One-shot-subgraph Link Prediction on Knowledge Graphs

---

**Require:** KG $\mathcal{G} = (\mathcal{V}, \mathcal{R}, \mathcal{E})$, degree matrix $\boldsymbol{D} \in \mathbb{R}^{|\mathcal{V}| \times |\mathcal{V}|}$, adjacency matrix $\boldsymbol{A} \in \{0, 1\}^{|\mathcal{V}| \times |\mathcal{V}|}$, damping coefficient $\alpha$, maximun PPR iterations $K$, query $(u, q, ?)$, sampler $g_\phi$, predictor $f_\theta$.

1: # Step-1. Generate sampling distribution
2: initialize $\boldsymbol{s} \leftarrow \mathbb{1}(u)$, $\boldsymbol{p}^{(0)} \leftarrow \mathbb{1}(u)$.
3: **for** $k = 1 \ldots K$ **do**
4:     $\boldsymbol{p}^{(k+1)} \leftarrow \alpha \cdot \boldsymbol{s} + (1 - \alpha) \cdot \boldsymbol{D}^{-1} \boldsymbol{A} \cdot \boldsymbol{p}^{(k)}$.
5: **end for**
6: # Step-2. Extract a subgraph $\mathcal{G}_s$
7: $\mathcal{V}_s \leftarrow \texttt{TopK}(\mathcal{V}, \boldsymbol{p}, K = r_\mathcal{V}^q \times |\mathcal{V}|)$.
8: $\mathcal{E}_s \leftarrow \texttt{TopK}(\mathcal{E}, \{\boldsymbol{p}_u \cdot \boldsymbol{p}_v : u, v \in \mathcal{V}_s, (u, r, v) \in \mathcal{E}\}, K = r_\mathcal{E}^q \times |\mathcal{E}|)$.
9: # Step-3. Reason on the subgraph
10: initialize representations $\boldsymbol{h}_o^{(0)} \leftarrow \mathbb{1}(o = u)$.
11: **for** $\ell = 1 \ldots L$ **do**
12:     $\boldsymbol{h}_o^{(\ell)} \leftarrow \texttt{DROPOUT}(\texttt{ACT}(\texttt{AGG}\{\texttt{MESS}(\boldsymbol{h}_x^{(\ell-1)}, \boldsymbol{h}_r^{(\ell-1)}, \boldsymbol{h}_o^{(\ell-1)}) : (x, r, o) \in \mathcal{E}_s\}))$.
13: **end for**
14: **return** Prediction $\hat{\boldsymbol{y}}_o = \texttt{Readout}(\boldsymbol{h}_o^{(L)}, \boldsymbol{h}_u^{(L)})$ for each entity $o \in \mathcal{V}_s$.

---

### 4.2 OPTIMIZATION: EFFICIENT SEARCHING FOR DATA-ADAPTIVE CONFIGURATIONS

**Search space.** Note that hyperparameters $(r_\mathcal{V}^q, r_\mathcal{E}^q)$ and $L$ play important roles in Algorithm 1. Analytically, a larger subgraph with larger $r_\mathcal{V}^q, r_\mathcal{E}^q$ does not indicate a better performance, as more irrelevant information is also covered. Besides, a deeper model with a larger $L$ can capture more complex patterns but is more likely to suffer from the over-smoothing problem (Oono & Suzuki, 2019). Overall, the $(r_\mathcal{V}^q, r_\mathcal{E}^q)$ are for sampler's hyper-parameters $\phi_{\text{hyper}}$. In addition to $L$, predictor's hyper-parameters $\theta_{\text{hyper}}$ contain several intra-layer or inter-layer designs, as illustrated in Fig. 2(a).

**Search problem.** Next, we propose the bi-level optimization problem to adaptively search for the optimal configuration $(\phi_{\text{hyper}}^*, \theta_{\text{hyper}}^*)$ of design choices on a specific KG, namely,

$$\phi_{\text{hyper}}^* = \arg \max_{\phi_{\text{hyper}}} \mathcal{M}(f_{(\theta_{\text{hyper}}^*, \theta_{\text{learn}}^*)}, g_{\phi_{\text{hyper}}}, \mathcal{E}^{\text{val}}),$$

$$\text{s.t. } \theta_{\text{hyper}}^* = \arg \max_{\theta_{\text{hyper}}} \mathcal{M}(f_{(\theta_{\text{hyper}}, \theta_{\text{learn}}^*)}, g_{\bar{\phi}_{\text{hyper}}}, \mathcal{E}^{\text{val}}), \quad (5)$$

where the performance measurement $\mathcal{M}$ can be Mean Reciprocal Ranking (MRR) or Hits@k. Note the non-parametric sampler $g_\phi$ only contains hyper-parameters $\phi_{\text{hyper}}$. As for predictor $f_\theta$, its $\theta = (\theta_{\text{hyper}}, \theta_{\text{learn}})$ also includes learnable $\theta_{\text{learn}}$ that $\theta_{\text{learn}}^* = \arg \min_{\theta_{\text{learn}}} \mathcal{L}_{cls}(f_{(\theta_{\text{hyper}}, \theta_{\text{learn}})}, g_{\bar{\phi}_{\text{hyper}}}, \mathcal{E}^{\text{train}})$.

**Search algorithm.** Directly searching on both data and model spaces is expensive due to the large space size and data scale. Hence, we split the search into two sub-processes as Fig. 2(b), *i.e.*,

- First, we freeze the sampler $g_{\bar{\phi}}$ (with constant $\phi_{\text{hyper}}$) to search for the optimal predictor $f_{\theta^*}$ with (1) the hyper-parameters optimization for $\theta_{\text{hyper}}^*$ and (2) the stochastic gradient descent for $\theta_{\text{learn}}^*$.

- Then, we freeze the predictor $f_{\theta^*}$ and search for the optimal sampler $g_{\phi^*}$, simplifying to pure hyper-parameters optimization for $\phi_{\text{hyper}}^*$ in a zero-gradient manner with low computation complexity.

Specifically, we follow the sequential model-based Bayesian Optimization (BO) (Bergstra et al., 2013; Hutter et al., 2011) to obtain $\phi_{\text{hyper}}^*$ and $\theta_{\text{hyper}}^*$. Random forest (RF) (Breiman, 2001) is chosen as the surrogate model because it has a stronger power for approximating the complex and discrete

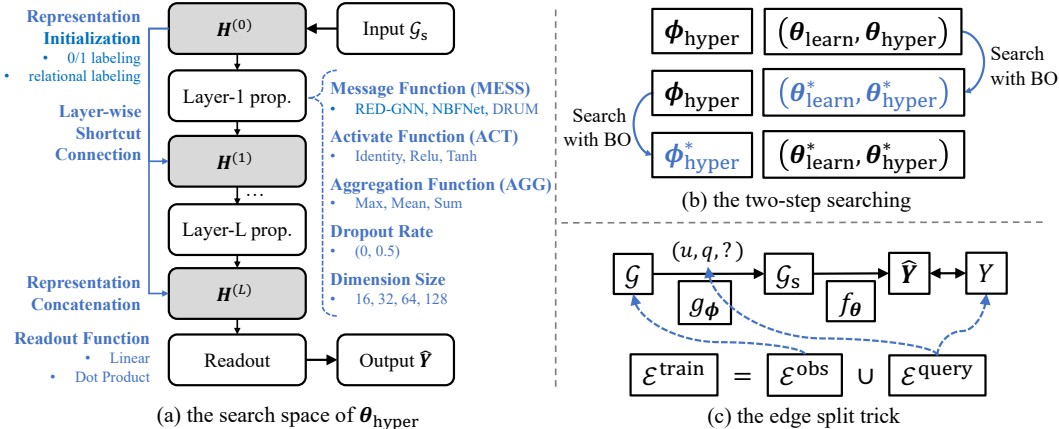

Figure 2: Illustrations of the optimization procedure. Note the predictor in (a) is with hyper-parameters and learnable parameters in each layer's propagation, and the $H^{(L)}$ indicates the representations in $L$-th layer. By contrast, the sampler only contains hyper-parameters as it does not require learning.

curvature (Grinsztajn et al., 2022), compared with other common surrogates, *e.g.*, Gaussian Process (GP) (Williams & Rasmussen, 1995) or Multi-layer Perceptron (MLP) (Gardner & Dorling, 1998).

**Acceleration in searching.** We adopt a data split trick that balances observations and predictions. It saves time as the training does not necessarily traverse all the training facts. Specifically, partial training facts are sampled as training queries while the others are treated as observations, *i.e.*, we randomly separate the training facts into two parts as $\mathcal{E}^{\text{train}} = \mathcal{E}^{\text{obs}} \cup \mathcal{E}^{\text{query}}$, where the overall prediction system $f_{\boldsymbol{\theta}} \circ g_{\boldsymbol{\phi}}$ takes $\mathcal{E}^{\text{obs}}$ as input and then predicts $\mathcal{E}^{\text{query}}$ (see Fig. 2(c)). Here, the split ratio $r^{\text{split}}$ is to balance the sizes of these two parts as $r^{\text{split}} = |\mathcal{E}^{\text{obs}}|/|\mathcal{E}^{\text{query}}|$. Thus, the training becomes $\boldsymbol{\theta}^* = \arg\min_{\boldsymbol{\theta}} \Sigma_{(u,q,v)\in\mathcal{E}^{\text{query}}} \mathcal{L}_{cls}(f_{\boldsymbol{\theta}}(\mathcal{G}_s), v)$ with the split query edges $\mathcal{E}^{\text{query}}$, where $\mathcal{G}_s = g_{\boldsymbol{\phi}}(\mathcal{E}^{\text{obs}}, u, q)$ with the split observation edges $\mathcal{E}^{\text{obs}}$. More technical details can be found in the Appendix. B.

### 4.3 THEORY: THE EXTRAPOLATION POWER OF ONE-SHOT-SUBGRAPH LINK PREDICTION

Further, we investigate the extrapolation power of link prediction across graph scales, *i.e.*, training and inference on different scales of graphs. For example, training on small subgraphs $\mathcal{G}_s^{\text{train}}$ and testing on large subgraphs $\mathcal{G}_s^{\text{test}}$, where the ratio of entities $|\mathcal{V}_s^{\text{test}}|/|\mathcal{V}_s^{\text{train}}| \gg 1$. This scenario is practical for recommendation systems on social networks that can encounter much larger graphs in the testing phase. Intuitively, training on smaller $\mathcal{G}_s^{\text{train}}$ can save time for its faster convergence on subgraphs (Shi et al., 2023), while predicting on larger $\mathcal{G}_s^{\text{test}}$ might gain promotion for more support of facts in $\mathcal{G}_s^{\text{test}}$.

Nonetheless, the Theorem 1 below proves that link prediction can become unreliable as the test graph grows. That is, if we use a *small* subgraph for training and a *large* subgraph for testing, the predictor will struggle to give different predictions within and across sampling distributions by $g$, even when these probabilities are arbitrarily different in the underlying graph model. Our empirical results in Fig. 4 support this theoretical finding. Hence, it is necessary to strike a balance of subgraphs' scale.

**Theorem 1.** *Let $\mathcal{G}_s^{train} \sim \mathbb{P}_{\mathcal{G}}$ and $\mathcal{G}_s^{test} \sim \mathbb{P}_{\mathcal{G}}$ be the training and testing graphs that are sampled from distribution $\mathbb{P}_{\mathcal{G}}$. Consider any two test entities $u, v \in \mathcal{V}_s^{test}$, for which we can make a prediction decision of fact $(u, q, v)$ with the predictor $f_{\boldsymbol{\theta}}$, i.e., $\hat{\boldsymbol{y}}_v = f_{\boldsymbol{\theta}}(\mathcal{G}_s^{test})_v \neq \tau$. Let $\mathcal{G}^{test}$ be large enough to satisfy $\sqrt{|\mathcal{V}_s^{test}|}/\sqrt{\log(2|\mathcal{V}_s^{test}|/p)} \geq {}^{4\sqrt{2}}/d_{\min}$, where $d_{\min}$ is the constant of graphon degree (Diaconis & Janson, 2007). Then, for an arbitrary threshold $\tau \in [0, 1]$, the testing subgraph $\mathcal{G}_s^{test}$ satisfies that*

$$\frac{\sqrt{|\mathcal{V}_s^{test}|}}{\sqrt{\log(2|\mathcal{V}_s^{test}|/p)}} \geq \frac{2(C_1 + C_2\|g\|_\infty)}{|f_{\boldsymbol{\theta}}(\mathcal{G}_s^{test})_v - \tau|/L(M^{train})}. \tag{6}$$

*where the underlying generative function of graph signal $g \in L^\infty$ is with the essential supreme norm as in (Maskey et al., 2022; Zhou et al., 2022). The $p, C_1, C_2$ are constants and depend on $M^{train}$ where $\min(supp(|\mathcal{V}_s^{test}|)) \gg M^{train} = \max(supp(|\mathcal{V}_s^{train}|))$. It means any test graph can be much larger than the largest possible training graph, and $supp$ indicates the support of a distribution. Then, if $u$ and $v$ are isomorphic in topology and with the same representations, we have a probability at least $1 - \sum_{\ell=1}^{L} 2(|\boldsymbol{h}^{(\ell)}| + 1)p$ with hidden size $|\boldsymbol{h}^{(\ell)}|$ that the same predictions can be obtained whether $u, v$ are generated by the same or distinct $g$. The detailed proof can be found in Appendix. A.*

Table 1: Empirical results of WN18RR, NELL-995, YAGO3-10 datasets. Best performance is indicated by the **bold face** numbers, and the underline means the second best. "–" means unavailable results. "H@1" and "H@10" are short for Hit@1 and Hit@10 (in percentage), respectively.

| type | models | WN18RR | | | NELL-995 | | | YAGO3-10 | | |
|------|--------|--------|--------|--------|--------|--------|--------|--------|--------|--------|
| | | MRR↑ | H@1↑ | H@10↑ | MRR↑ | H@1↑ | H@10↑ | MRR↑ | H@1↑ | H@10↑ |
| Semantic Models | ConvE | 0.427 | 39.2 | 49.8 | 0.511 | 44.6 | 61.9 | 0.520 | 45.0 | 66.0 |
| | QuatE | 0.480 | 44.0 | 55.1 | 0.533 | 46.6 | 64.3 | 0.379 | 30.1 | 53.4 |
| | RotatE | 0.477 | 42.8 | 57.1 | 0.508 | 44.8 | 60.8 | 0.495 | 40.2 | 67.0 |
| Structural Models | MINERVA | 0.448 | 41.3 | 51.3 | 0.513 | 41.3 | 63.7 | – | – | – |
| | DRUM | 0.486 | 42.5 | 58.6 | 0.532 | 46.0 | **66.2** | 0.531 | 45.3 | 67.6 |
| | RNNLogic | 0.483 | 44.6 | 55.8 | 0.416 | 36.3 | 47.8 | 0.554 | 50.9 | 62.2 |
| | CompGCN | 0.479 | 44.3 | 54.6 | 0.463 | 38.3 | 59.6 | 0.489 | 39.5 | 58.2 |
| | DPMPN | 0.482 | 44.4 | 55.8 | 0.513 | 45.2 | 61.5 | 0.553 | 48.4 | 67.9 |
| | NBFNet | 0.551 | 49.7 | **66.6** | 0.525 | 45.1 | 63.9 | 0.550 | 47.9 | 68.3 |
| | RED-GNN | 0.533 | 48.5 | 62.4 | 0.543 | 47.6 | 65.1 | 0.559 | 48.3 | 68.9 |
| | **one-shot-subgraph** | **0.567** | **51.4** | **66.6** | **0.547** | **48.5** | 65.1 | **0.606** | **54.0** | **72.1** |

Table 2: Empirical results of two OGB datasets (Hu et al., 2020) with regard to official leaderboards.

| type | models | OGBL-BIOKG | | | OGBL-WIKIKG2 | | |
|------|--------|-----------|-----------|-----------|-----------|-----------|-----------|
| | | Test MRR↑ | Valid MRR↑ | #Params↓ | Test MRR↑ | Valid MRR↑ | #Params↓ |
| Semantic Models | TripleRE | 0.8348 | 0.8360 | 469,630,002 | 0.5794 | 0.6045 | 500,763,337 |
| | AutoSF | 0.8309 | 0.8317 | 93,824,000 | 0.5458 | 0.5510 | 500,227,800 |
| | PairRE | 0.8164 | 0.8172 | 187,750,000 | 0.5208 | 0.5423 | 500,334,800 |
| | ComplEx | 0.8095 | 0.8105 | 187,648,000 | 0.4027 | 0.3759 | 1,250,569,500 |
| | DistMult | 0.8043 | 0.8055 | 187,648,000 | 0.3729 | 0.3506 | 1,250,569,500 |
| | RotatE | 0.7989 | 0.7997 | 187,597,000 | 0.4332 | 0.4353 | 1,250,435,750 |
| | TransE | 0.7452 | 0.7456 | 187,648,000 | 0.4256 | 0.4272 | 1,250,569,500 |
| Structural Models | **one-shot-subgraph** | **0.8430** | **0.8435** | **976,801** | **0.6755** | **0.7080** | **6,831,201** |

Table 3: Coverage Ratio of different heuristics. **Bold face** numbers indicate the best results in column.

| heuristics | WN18RR | | | NELL-995 | | | YAGO3-10 | | |
|------------|--------|--------|--------|--------|--------|--------|--------|--------|--------|
| | $r_{\mathcal{V}}^q=0.1$ | $r_{\mathcal{V}}^q=0.2$ | $r_{\mathcal{V}}^q=0.5$ | $r_{\mathcal{V}}^q=0.1$ | $r_{\mathcal{V}}^q=0.2$ | $r_{\mathcal{V}}^q=0.5$ | $r_{\mathcal{V}}^q=0.1$ | $r_{\mathcal{V}}^q=0.2$ | $r_{\mathcal{V}}^q=0.5$ |
| Random Sampling (RAND) | 0.100 | 0.200 | 0.500 | 0.100 | 0.200 | 0.500 | 0.100 | 0.200 | 0.500 |
| PageRank (PR) | 0.278 | 0.407 | 0.633 | 0.405 | 0.454 | 0.603 | 0.340 | 0.432 | 0.694 |
| Random Walk (RW) | 0.315 | 0.447 | 0.694 | 0.522 | 0.552 | 0.710 | 0.449 | 0.510 | 0.681 |
| Breadth-first-searching (BFS) | 0.818 | 0.858 | 0.898 | 0.872 | 0.935 | 0.982 | 0.728 | 0.760 | 0.848 |
| Personalized PageRank (PPR) | **0.876** | **0.896** | **0.929** | **0.965** | **0.977** | **0.987** | **0.943** | **0.957** | **0.973** |

## 5 EXPERIMENTS

In this section, we empirically verify the effectiveness of the proposed framework. The major experiments are conducted with PyTorch (Paszke et al., 2017) and one NVIDIA RTX 3090 GPU. The OGB datasets are run with one NVIDIA A100 GPU. We use five benchmarks with more than ten thousand entities (see Tab. 11), including WN18RR (Dettmers et al., 2017), NELL-995 (Xiong et al., 2017), YAGO3-10 (Suchanek et al., 2007), OGBL-BIOKG, and OGBL-WIKIKG2 (Hu et al., 2020).

**Metrics.** We adopt the filtered ranking-based metrics for evaluation, *i.e.*, mean reciprocal ranking (MRR) and Hit@$k$ (*i.e.*, both Hit@1 and Hit@10), following (Bordes et al., 2013; Teru et al., 2020; Wang et al., 2017; Zhu et al., 2021). For both metrics, a higher value indicates a better performance.

**Main Results.** As results shown in Tab. 1 and Tab. 2, our one-shot-subgraph link prediction method achieves leading performances on all five large-scale benchmarks over all the baselines. Especially on the largest OGBL-WIKIKG2 dataset, a $16.6\%$ improvement in Test MRR can be achieved. Note the results attribute to *a deep GNN (high expressiveness) and small subgraphs (essential information)* extracted by sampling $10\%$ of entities on average for answering specific queries. Which means, it is *unnecessary* to utilize the whole KG in link prediction; meanwhile, only a small proportion of entities and facts are essential for answering specific queries that can be quickly identified by the PPR heuristics. In what follows, we conduct an in-depth analysis of the properties of the proposed method.

**The Sampling Distribution.** We empirically evaluate to what extent the entities relevant to a specific query can be identified by heuristics, *e.g.*, BFS, RW, and PPR. We quantify their power of identifying potential answers via the metric of Coverage Ratio $\mathrm{CR}=\frac{1}{|\mathcal{E}^{\text{test}}|}\sum_{(u,q,v)\in\mathcal{E}^{\text{test}}}\mathbb{I}\{v\in\mathcal{V}_s\}$, *i.e.*, the ratio of covered answer entities that remain in the set of sampled entities $\mathcal{V}_s$. As shown in Tab. 3 and Fig. 3, PPR gets a much higher CR and notably outperforms other heuristics in identifying potential answers.

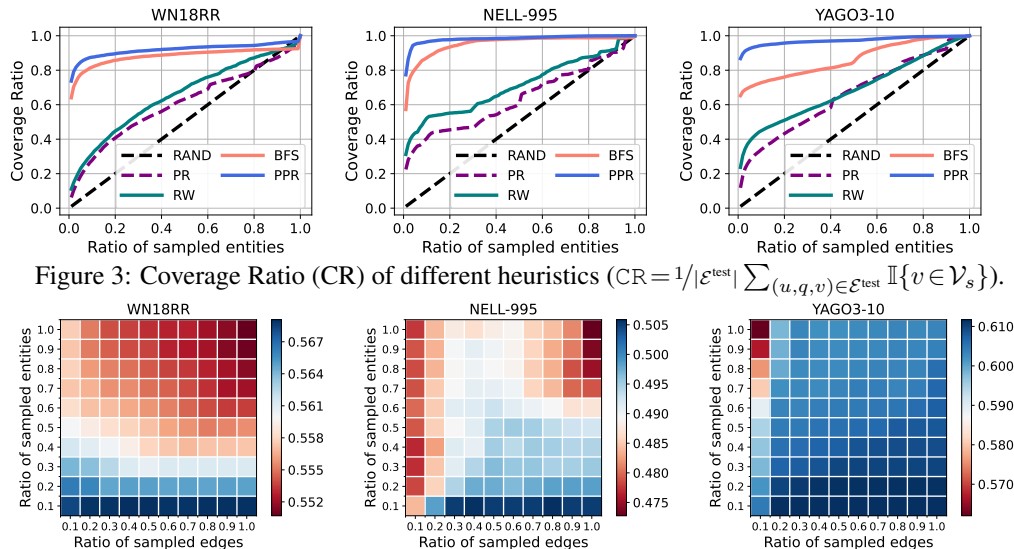

Figure 3: Coverage Ratio (CR) of different heuristics ($\mathrm{CR} = 1/|\mathcal{E}^{\text{test}}| \sum_{(u,q,v) \in \mathcal{E}^{\text{test}}} \mathbb{I}\{v \in \mathcal{V}_s\}$).

Figure 4: Heatmaps of validate MRR (the higher, the better) *w.r.t.* $r_{\mathcal{V}}^q$ and $r_{\mathcal{E}}^q$ on three benchmarks.

Table 4: Comparison of effectiveness with regard to subgraph sampling.

| #layers ($L$) | $r_{\mathcal{V}}^q$ | $r_{\mathcal{E}}^q$ | WN18RR | | | NELL-995 | | | YAGO3-10 | | |
|---|---|---|---|---|---|---|---|---|---|---|---|
| | | | MRR | H@1 | H@10 | MRR | H@1 | H@10 | MRR | H@1 | H@10 |
| 8 | 1.0 | 1.0 | Out of memory | | | Out of memory | | | Out of memory | | |
| 8 | 0.1 | 1.0 | **0.567** | **51.4** | **66.6** | **0.547** | **48.5** | **65.1** | **0.606** | **54.0** | **72.1** |
| 6 | 1.0 | 1.0 | 0.543 | 49.2 | 64.3 | 0.519 | 45.3 | 62.7 | 0.538 | 46.9 | 66.0 |
| 6 | 0.1 | 1.0 | 0.566 | 51.2 | 66.5 | 0.540 | 48.0 | 63.8 | 0.599 | 53.1 | 71.8 |
| 4 | 1.0 | 1.0 | 0.513 | 46.6 | 59.8 | 0.518 | 45.4 | 61.5 | 0.542 | 47.6 | 66.1 |
| 4 | 0.1 | 1.0 | 0.523 | 47.7 | 60.5 | 0.538 | 47.4 | 63.4 | 0.589 | 52.2 | 70.4 |

**Ablation Study. (1) Training with varying scales and layers:** We train the predictor from scratch with various scales of subgraphs and the number of layers. As can be seen from Tab. 4, involving all the entities with $r_{\mathcal{V}}^q = 1.0$ degenerates the prediction, as too many irrelevant entities are covered. Besides, a deeper predictor with a larger $L$ consistently brings better results. These observations enlighten us to learn a deeper predictor with small subgraphs. **(2) Training with different heuristics:** We replace the PPR sampling module with four other common heuristics. However, as shown in Tab. 5, their final prediction performances are outperformed by PPR. **(3) Test-time extrapolation power across scales:** As in Theorem. 1, we evaluate the extrapolation power by generalizing to various scales of subgraphs that are different from the scale of training graphs, *e.g.*, the whole graph $r_{\mathcal{V}}^q = r_{\mathcal{E}}^q = 1.0$. As shown in Fig. 4, the predictor also suffers when prediction with more irrelevant entities, especially with larger $r_{\mathcal{V}}^q$, while the generally good cases are a lower $r_{\mathcal{V}}^q$ to focus on the relevant entities and a high $r_{\mathcal{E}}^q$ to preserve the local structure (of head $u$) within the sampled entities.

**Training and Inference Efficiency.** Next, we conduct an efficiency study to investigate the improvement of efficiency brought by the proposed one-shot-subgraph link prediction framework. The running time and GPU memory of an 8-layer GNN are summarized in Tab. 7. As can be seen, a notable advantage of decoupling is that it has less computing cost in both terms of less running time and also less memory cost. Particularly, on the YAGO3-10 dataset, the existing GNN-based methods will run out of memory with a deep architecture. However, with the subgraph sampling of lower ratios of $r_{\mathcal{V}}^q$ and $r_{\mathcal{E}}^q$, the learning and prediction of GNNs become feasible that is with less memory cost and achieving state-of-the-art performance. Hence, we show that our method is effective and also efficient that it supports the learning and prediction of deep GNNs on large-scale knowledge graphs.

Besides, we provide a detailed efficiency comparison between our method (with 10% entities) and the original implementation (with 100% entities) on two SOTA methods, NBFNet and RED-GNN. Tab. 6 shows that the training time is significantly reduced when learning with our method. Notably, on the YAGO3-10 dataset, $94.3\%$ and $94.5\%$ of training time (for one epoch) can be saved for NBFNet and RED-GNN, respectively. Besides, our method boosts the performance as advantage 2, where the performance improvement can come from a deeper GNN and a smaller observation graph (detailed analysis in Appendix. D.3). Full evaluations and more discussions are elaborated in the Appendix. C.

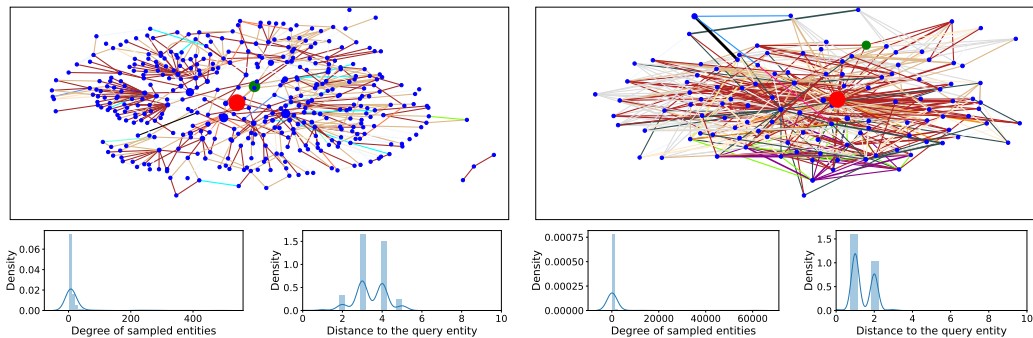

Figure 5: Exemplar subgraphs sampled from WN18RR (left) and YAGO3-10 (right). The red and green nodes indicate the query entity and answer entity. The colors of the edges indicate relation types. The bottom distributions of degree and distance show the statistical properties of each subgraph.

Table 5: Comparison of prediction performance with different sampling heuristics.

| heuristics | WN18RR | | | YAGO3-10 | | |
|---|---|---|---|---|---|---|
| | MRR | H@1 | H@10 | MRR | H@1 | H@10 |
| Random Sampling (RAND) | 0.03 | 43.4 | 3.5 | 0.057 | 5.1 | 6.5 |
| PageRank (PR) | 0.124 | 11.5 | 14.2 | 0.315 | 28.9 | 35.9 |
| Random Walk (RW) | 0.507 | 45.8 | 59.8 | 0.538 | 46.3 | 67.2 |
| Breadth-first-searching (BFS) | 0.543 | 49.6 | 63.0 | 0.562 | 49.4 | 69.0 |
| **Personalized PageRank (PPR)** | **0.567** | **51.4** | **66.6** | **0.606** | **54.0** | **72.1** |

Table 6: Comparison of prediction performance with two recent GNN methods.

| methods | WN18RR | | | | YAGO3-10 | | | |
|---|---|---|---|---|---|---|---|---|
| | MRR | H@1 | H@10 | Time | MRR | H@1 | H@10 | Time |
| NBFNet (100% entities) | 0.551 | 49.7 | 66.6 | 32.3 min | 0.550 | 47.9 | 68.3 | 493.8 min |
| NBFNet + one-shot-subgraph (10% entities) | **0.554** | **50.5** | **66.3** | **2.6 min** | **0.565** | **49.6** | **69.2** | **28.2 min** |
| RED-GNN (100% entities) | 0.533 | 48.5 | 62.4 | 68.7 min | 0.559 | 48.3 | 68.9 | 1382.9 min |
| RED-GNN + one-shot-subgraph (10% entities) | 0.567 | 51.4 | 66.6 | 4.5 min | 0.606 | 54.0 | 72.1 | 76.3 min |

Table 7: Comparison of efficiency with an 8-layer predictor and different $r_{\mathcal{V}}^q, r_{\mathcal{E}}^q$.

| phase | $r_{\mathcal{V}}^q$ | $r_{\mathcal{E}}^q$ | WN18RR | | NELL-995 | | YAGO3-10 | |
|---|---|---|---|---|---|---|---|---|
| | | | Time | Memory | Time | Memory | Time | Memory |
| Training | 1.0 | 1.0 | Out of memory | | Out of memory | | Out of memory | |
| | 0.5 | 0.5 | 26.3m | 20.3GB | 1.6h | 20.1GB | Out of memory | |
| | 0.2 | 1.0 | 12.8m | 20.2GB | 1.2h | 18.5GB | Out of memory | |
| | 0.2 | 0.2 | 6.7m | 6.4GB | 0.6h | 8.9GB | 2.1h | 23.1GB |
| | 0.1 | 1.0 | 7.2m | 9.8GB | 0.8h | 12.1GB | 1.3h | 13.9GB |
| | 0.1 | 0.1 | 6.6m | 5.1GB | 0.3h | 5.3GB | 0.9h | 10.2GB |
| Inference | 1.0 | 1.0 | 7.3m | 6.7GB | 17.5m | 12.8GB | 1.6h | 15.0GB |
| | 0.5 | 0.5 | 6.0m | 4.3GB | 8.3m | 4.5GB | 1.1h | 10.1GB |
| | 0.2 | 1.0 | 3.2m | 5.8GB | 4.2m | 12.1GB | 0.7h | 14.7GB |
| | 0.2 | 0.2 | 2.8m | 1.9GB | 3.6m | 2.5GB | 0.6h | 3.7GB |
| | 0.1 | 1.0 | 2.7m | 2.7GB | 3.1m | 9.4GB | 0.4h | 9.7GB |
| | 0.1 | 0.1 | 2.3m | 1.7GB | 2.9m | 1.9GB | 0.4h | 3.1GB |

**Case Study.** We visualize the sampled subgraph in Fig. 5 with the histograms of degree and distance distributions. As can be seen, the local structure of query entity $u$ is well preserved, while the true answers $v$ are also covered in the subgraphs. More cases and analyses can be found in Appendix. E.

## 6 CONCLUSION

In this paper, we propose the one-shot-subgraph link prediction to alleviate the scalability problem of structural methods and achieve efficient as well as adaptive learning on large-scale KGs. We discover that the non-parametric and computation-efficient heuristics PPR can effectively identify the potential answers and support to the prediction. We further introduce the automated searching for adaptive configurations in both data space and model space. Extensive experiments on five large-scale benchmarks verify the effectiveness and efficiency of our method. Importantly, we show it unnecessary to utilize the whole KG for answering specific queries; meanwhile, only a small proportion of information is essential and can be identified by the PPR heuristics without learning.

## ACKNOWLEDGMENTS

ZKZ and BH were supported by the NSFC General Program No. 62376235, Guangdong Basic and Applied Basic Research Foundation Nos. 2022A1515011652 and 2024A1515012399, HKBU Faculty Niche Research Areas No. RC-FNRA-IG/22-23/SCI/04, and HKBU CSD Departmental Incentive Scheme. JCY was supported by 111 plan (No. BP0719010) and National Natural Science Foundation of China (No. 62306178). QMY was in part supported by NSFC (No. 92270106) and National Key Research and Development Program of China under Grant 2023YFB2903904. The authors thank Haobo Xu for his assistance in experiments.

## ETHICS STATEMENT

We would claim that this work does not raise any ethical concerns. Besides, this work does not involve any human subjects, practices to data set releases, potentially harmful insights, methodologies and applications, potential conflicts of interest and sponsorship, discrimination/bias/fairness concerns, privacy and security issues, legal compliance, and research integrity issues.

## REPRODUCIBILITY STATEMENT

The experimental setups for training and evaluation are described in detail in Sec. 5 and Appendix. C, and the experiments are all conducted using public datasets. The code is publicly available at: https://github.com/tmlr-group/one-shot-subgraph.

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

# Appendix

## Table of Contents

# A   THEORETICAL ANALYSIS

## A.1   NOTATIONS

We summarize the frequently used notations in Tab. 8.

Table 8: The most frequently used notations in this paper.

| notations | meanings |
|---|---|
| $\mathcal{V}, \mathcal{R}, \mathcal{E}$ | the set of entities, relations, facts (edges) of the original KG |
| $\mathcal{G} = (\mathcal{V}, \mathcal{R}, \mathcal{E})$ | the original KG |
| $\mathcal{V}_s, \mathcal{R}_s, \mathcal{E}_s$ | the set of entities, relations, facts (edges) of the sampled KG |
| $\mathcal{G}_s = (\mathcal{V}_s, \mathcal{R}_s, \mathcal{E}_s)$ | the sampled KG generated by the sampler $g_\phi$ |
| $(x, r, o)$ | a fact triplet in $\mathcal{E}$ with subject entity $x$, relation $r$ and object entity $o$ |
| $(u, q, v)$ | a query triple with query entity $u$, query relation $q$ and answer entity $v$ |
| $L$ | the total number of propagation steps |
| $\ell$ | the $\ell$-th propagation step and $\ell \in \{0 \ldots L\}$ |
| $K$ | the maximum steps of updating the PPR scores |
| $k$ | the $k$-th propagation step and $k \in \{0 \ldots K\}$ |
| $\boldsymbol{h}_o^{(\ell)}$ | the representation of entity $o$ at step $\ell$ |
| $g_\phi$ | the sampler of the one-shot-subgraph link prediction framework |
| $f_{\boldsymbol{\theta}}$ | the predictor of the one-shot-subgraph link prediction framework |
| $r_{\mathcal{V}}^q, r_{\mathcal{E}}^q$ | the sampling ratios of entities and edges |

## A.2   COMPLEXITY ANALYSIS

Next, we compare three different manners from perspectives of computation and parameter.

In a nutshell, semantic models are computation-efficient but parameter-expensive, while structural models are parameter-efficient but computation-expensive. Here, we aim to make the best of the both worlds by designing a framework that is parameter-efficient and computation-efficient. The proposed one-shot-subgraph link prediction models, as will be elaborated in Sec. 3, is with the new prediction manner that $\mathcal{G} \xmapsto{g_\phi, (u_q, r_q)} \mathcal{G}_s \xmapsto{f_{\boldsymbol{\theta}}} \hat{\boldsymbol{Y}}$. The key difference here is that, instead of the original $\mathcal{G}$, the predictor $f_{\boldsymbol{\theta}}$ is acting on the query-dependent subgraph $\mathcal{G}_s$ extracted by the sampler $g_\phi$.

By contrast, we show that the proposed subgraph models make the best of both worlds: (1) subgraph models are computation-efficient, as the extracted subgraph is much smaller than the original graph, *i.e.*, $|\mathcal{V}_s| \ll |\mathcal{V}|$ and $|\mathcal{E}_s| \ll |\mathcal{E}|$; (2) inherits from Sadeghian et al. (2019); Zhu et al. (2021); Zhang & Yao (2022), subgraph models are also parameter-efficient: only requires the relations' embeddings but not the expensive entities' embeddings. By contrast, the semantic models need to learn entities's embeddings, which are parameter-expensive and only applicable in transductive settings.

A detailed comparison of parameter and computation complexity is summarized in Tab. 9.

Table 9: The comparison of related works with parameter complexity and computation complexity.

| Category | Parameter complexity | Computation complexity |
|---|---|---|
| semantic models | $O(|\mathcal{V}| \cdot D_{\mathcal{V}}) + O(|\mathcal{R}| \cdot D_{\mathcal{R}})$ | $O(|\mathcal{R}| \cdot D_{\mathcal{V}})$ |
| structural models | $O(|\mathcal{R}| \cdot D_{\mathcal{R}})$ | $O(|\mathcal{E}| \cdot D_{\mathcal{R}} \cdot L)$ |
| subgraph models | $O(|\mathcal{R}| \cdot D_{\mathcal{R}})$ | $O(|\mathcal{E}| \cdot K) + O(|\mathcal{E}_s| \cdot D_{\mathcal{R}} \cdot L)$ |

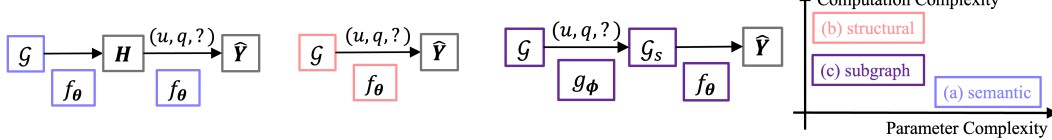

(a) semantic model     (b) structural model     (c) one-shot-subgraph model     (d) complexity comparison

Figure 6: Illustrations of prediction manners. Semantic and structure models implicitly or explicitly take the whole graph $\mathcal{G}$ for prediction. Our *one-shot-subgraph* model *only requires one subgraph $\mathcal{G}_s$ for the prediction of one query* that decreases computation complexity via decoupling $f_\theta$ and $\mathcal{G}$.

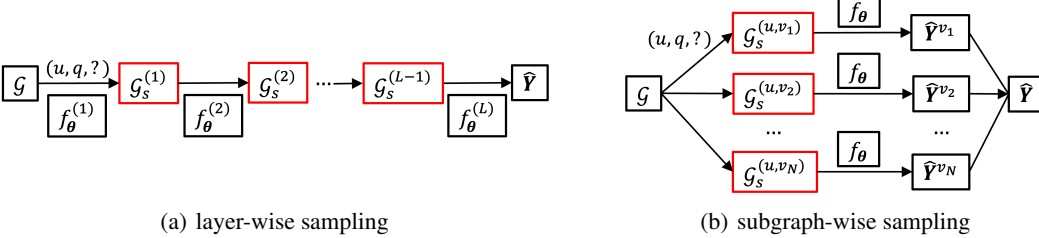

(a) layer-wise sampling            (b) subgraph-wise sampling

Figure 7: Illustrations of sampling methods. Detailed analysis is corresponding to Sec. 3.

### A.3 PROOF FOR THEOREM 1

*Proof.* Here, let $\mathcal{G}_s^{\text{train}} \sim \mathbb{P}_{\mathcal{G}}$ and $\mathcal{G}_s^{\text{test}} \sim \mathbb{P}_{\mathcal{G}}$ be the training and testing graphs that are sampled distribution $\mathbb{P}_{\mathcal{G}}$. Let $\mathcal{G}^{\text{test}}$ be large enough to satisfy $\sqrt{|\mathcal{V}_s^{\text{test}}|}/\sqrt{\log(2|\mathcal{V}_s^{\text{test}}|/p)} \geq 4\sqrt{2}/d_{\min}$, where $d_{\min}$ is the constant of graphon degree (Diaconis & Janson, 2007).

In this part, we follow Maskey et al. (2022); Zhou et al. (2022) with the definitions of graph message-passing neural network $\Phi$ and continuous message passing neural network $\Psi$. Note that $\Phi$ is equivalent to the message passing function of the predictor $f_{\boldsymbol{w}}$ that generates the entity representation $\boldsymbol{h}_v^{(\ell)}$ in each layer $\ell = 1, 2, \cdots, L$. Besides, $\Psi$ is the continuous counterpart of $\Phi$. It uses the continuous aggregation instead of the discrete one w.r.t. the discrete graph topology in $\Phi$.

With the two above functions, $\|\Phi\|_\infty, \|\Psi\|_\infty$ can be trained and determined by the original graph $\mathcal{G}^{\text{train}}$ and hyperparameters $r_{\mathcal{V}}^q, r_{\mathcal{E}}^q$ and $L$. Following Maskey et al. (2022); Zhou et al. (2022), let $p \in (0, 1/\sum_{\ell=1}^L 2|\boldsymbol{h}^{(\ell)}|+1)$ we with probability at least $1-\sum_{\ell=1}^L 2(|\boldsymbol{h}^{(\ell)}|+1)p$, the difference $\delta$ between $\|\Phi\|_\infty$ and $\|\Psi\|_\infty$ is bounded by

$$
\begin{aligned}
\delta := &\max_{i=1,2,\cdots,|\mathcal{V}_s^{\text{test}}|} \|\Phi_i - \Psi_i\|_\infty \\
\leq &\sum_{\ell=1}^L L_\Psi^{(\ell)}(M^{\text{train}}) \Big(2\sqrt{2}\frac{(L_\Phi^{(\ell)}(M^{\text{train}})\|g\|_\infty + \|\Phi^{(\ell)}\|_\infty)\sqrt{\log(2|\mathcal{V}_s^{\text{test}}|/p)}}{\sqrt{|\mathcal{V}_s^{\text{test}}|}}\Big) \\
&\times \prod_{\ell'=\ell+1}^L \Big((L_\Psi^{(\ell')}(M^{\text{train}}))^2 + 2(L_\Phi^{(\ell')}(M^{\text{train}}))^2(L_\Psi^{(\ell')}(M^{\text{train}}))^2\Big).
\end{aligned}
$$

Then, based on the proof for Lemma B.9 in (Maskey et al., 2022), we can derive $\|g\|_\infty \leq B_1^{(\ell)} + B_2^{(\ell)}\|g\|_\infty$, where $B_1^{(\ell)}, B_2^{(\ell)}$ are independent of $g$. Specifically,

$$
B_1^{(\ell)} = \sum_{k=1}^\ell \Big(L_\Psi^{(k)}(M^{\text{train}})\|\Phi^{(k)}\|_\infty + \|\Psi^{(k)}\|_\infty\Big) \times \prod_{k'=k+1}^\ell \Big(L_\Psi^{(k')}(M^{\text{train}})(1 + L_\Psi^{(k')}(M^{\text{train}}))\Big),
$$

$$
B_2^{(\ell)} = \prod_{k=1}^\ell L_\Psi^{(k)}(M^{\text{train}})\Big(1 + L_\Phi^{(k)}(M^{\text{train}})\Big).
$$

Then, we define constants $C_1, C_2$ as follows.

$$C_1 = \sum_{\ell=1}^{L} L_\Psi^{(\ell)}(M^{\text{train}})\Big(2\sqrt{2}(L_\Phi^{(\ell)}(M^{\text{train}})B_1^{(\ell)} + \|\Phi^{(\ell+1)}\|_\infty)\Big)$$

$$\times \prod_{\ell'=\ell+1}^{L} \Big((L_\Psi^{(\ell')}(M^{\text{train}}))^2 + 2(L_\Phi^{(\ell')}(M^{\text{train}}))^2(L_\Psi^{(\ell')}(M^{\text{train}}))^2\Big),$$

$$C_2 = \sum_{\ell=1}^{L} L_\Psi^{(\ell)}(M^{\text{train}})\Big(2\sqrt{2}L_\Phi^{(\ell)}(M^{\text{train}})B_2^{(\ell)}\Big)$$

$$\times \prod_{\ell'=\ell+1}^{L} \Big((L_\Psi^{(\ell')}(M^{\text{train}}))^2 + 2(L_\Phi^{(\ell')}(M^{\text{train}}))^2(L_\Psi^{(\ell')}(M^{\text{train}}))^2\Big).$$

This, we can derive the difference $\delta$ as

$$\delta := \max_{i=1,2,\cdots,|\mathcal{V}_s^{\text{test}}|} \|\Phi_i - \Psi_i\|_\infty \le (C_1 + C_2\|g\|_\infty)\frac{\sqrt{\log(2|\mathcal{V}_s^{\text{test}}|/p)}}{\sqrt{|\mathcal{V}_s^{\text{test}}|}},$$

where $C_1, C_2$ depends on $\{L_\Phi^{(\ell)}(M^{\text{train}})\}_{\ell=1}^{L}$ and $\{L_\Psi^{(\ell)}(M^{\text{train}})\}_{\ell=1}^{L}$ that

$$\min(supp(|\mathcal{V}_s^{\text{test}}|)) \gg M^{\text{train}} = \max(supp(|\mathcal{V}_s^{\text{train}}|)).$$

Then, consider any two test entities $u, v \in \mathcal{V}_s^{\text{test}}$, for which we can make a prediction decision of fact $(u, q, v)$, we have

$$\|\Phi_u - \Phi_v\|_\infty \le \|\Phi_u - \Psi_u\|_\infty + \|\Psi_u - \Phi_v\|_\infty$$

$$= \|\Phi_u - \Psi_u\|_\infty + \|\Psi_v - \Phi_v\|_\infty \le (C_1 + C_2\|g\|_\infty)\frac{\sqrt{\log(2|\mathcal{V}_s^{\text{test}}|/p)}}{\sqrt{|\mathcal{V}_s^{\text{test}}|}}.$$

The first inequality holds by the triangle inequality. In contrast, the second inequality holds since $\Psi_u = \Psi_v$. Note that $u$ and $v$ are isomorphic in the topology of the test graph (Zhou et al., 2022) and thus with the same representations.

Next, with probability at least $1 - \sum_{\ell=1}^{L} 2(|\boldsymbol{h}^{(\ell)}| + 1)p$, for arbitrary entity $v' \in \mathcal{V}_s^{\text{test}}$, we have

$$\|\Phi_v - \Phi_v'\|_\infty \le (C_1 + C_2\|g\|_\infty)\frac{\sqrt{\log(2|\mathcal{V}_s^{\text{test}}|/p)}}{\sqrt{|\mathcal{V}_s^{\text{test}}|}}.$$

Then, when the size of the test graph is satisfied and $\boldsymbol{p}_{uv} = f_{\boldsymbol{w}}(\mathcal{G}_s^{\text{test}})_{uv} = \texttt{READOUT}(\boldsymbol{h}_u^{(L)}, \boldsymbol{h}_v^{(L)})$, we have

$$\|\boldsymbol{p}_{uv} - \boldsymbol{p}_{uv'}\|_\infty \le L(M^{\text{train}})\|\Phi_v - \Phi_v'\|_\infty \le \|\boldsymbol{p}_{uv} - \tau\|_\infty.$$

Specifically, if $\boldsymbol{p}_{uv} \ge \tau$, we have

$$\boldsymbol{p}_{uv'} \ge \boldsymbol{p}_{uv} - |\boldsymbol{p}_{uv} - \boldsymbol{p}_{uv'}| \ge \boldsymbol{p}_{uv} - |\boldsymbol{p}_{uv} - \tau| = \boldsymbol{p}_{uv} - \boldsymbol{p}_{uv} + \tau = \tau.$$

if $\boldsymbol{p}_{uv} \le \tau$, we have

$$\boldsymbol{p}_{uv'} \le \boldsymbol{p}_{uv} + |\boldsymbol{p}_{uv} - \boldsymbol{p}_{uv'}| \le \boldsymbol{p}_{uv} + |\boldsymbol{p}_{uv} - \tau| = \boldsymbol{p}_{uv} + \tau - \boldsymbol{p}_{uv} = \tau.$$

Thus, whether $u, v$ are generated by the same or distinct $g$, where the underlying generative function of graph signal $g \in L^\infty$ is with the essential supreme norm as in Maskey et al. (2022); Zhou et al. (2022), we have a probability at least $1 - \sum_{\ell=1}^{L} 2(|\boldsymbol{h}^{(\ell)}| + 1)p$ that the same predictions are obtained. $\qquad\square$

## B  IMPLEMENTATION DETAILS

**MESS($\cdot$) and AGG($\cdot$) for GNN-based methods.** The two functions of the GNN-based methods are summarized in Tab.10. The main differences between different methods are the combinators for entity and relation representations on the edges, the operators on the different representations, and the attention weights. Recent works (Vashishth et al., 2019; Zhu et al., 2021) have shown that the different combinators and operators only have a slight influence on the performance. Compared with GraIL and RED-GNN, even though the message functions and aggregation functions are similar, their empirical performances are quite different, with different propagation patterns.

Table 10: Summary of the GNN functions for message propagation. The values in {} represent different operation choices that are tuned as hyper-parameters for different datasets.

| method | $\boldsymbol{m}_{(e_s,r,e_o)}^{(\ell)} := \text{MESS}(\cdot)$ | $\boldsymbol{h}_{e_o}^{(\ell)} := \text{AGG}(\cdot)$ |
|---|---|---|
| R-GCN (Schlichtkrull et al., 2018) | $\boldsymbol{W}_r^{(\ell)}\boldsymbol{h}_{e_s}^{(\ell-1)}$, where $\boldsymbol{W}_r$ depends on the relation $r$ | $\boldsymbol{W}_o^{(\ell)}\boldsymbol{h}_{e_o}^{(\ell-1)} + \sum_{e_o \in \mathcal{N}(e_s)} \frac{1}{c_{o,r}} \boldsymbol{m}_{(e_s,r,e_o)}^{(\ell)}$ |
| CompGCN (Vashishth et al., 2019) | $\boldsymbol{W}_{\lambda(r)}^{(\ell)}\{-,*,\star\}(\boldsymbol{h}_{e_s}^{(\ell-1)}, \boldsymbol{h}_r^{(\ell)})$, where $\boldsymbol{W}_{\lambda(r)}^{(\ell)}$ depends on the direction of $r$ | $\sum_{e_o \in \mathcal{N}(e_s)} \boldsymbol{m}_{(e_s,r,e_o)}^{(\ell)}$ |
| GraIL (Teru et al., 2020) | $\alpha_{(e_s,r,e_o)\mid r_q}^{(\ell)}(\boldsymbol{W}_1^{(\ell)}\boldsymbol{h}_{e_s}^{(\ell-1)} + \boldsymbol{W}_2^{(\ell)}\boldsymbol{h}_{e_o}^{(\ell-1)})$, where $\alpha_{(e_s,r,e_o)\mid r_q}^{(\ell)}$ is the attention weight. | $\boldsymbol{W}_o^{(\ell)}\boldsymbol{h}_{e_s}^{(\ell-1)} + \sum_{e_o \in \mathcal{N}(e_s)} \frac{1}{c_{o,r}} \boldsymbol{m}_{(e_s,r,e_o)}^{(\ell)}$ |
| NBFNet (Zhu et al., 2021) | $\boldsymbol{W}^{(\ell)}\{+,*,\circ\}(\boldsymbol{h}_{e_s}^{(\ell-1)}, \boldsymbol{w}_q(r,r_q))$, where $\boldsymbol{w}_q(r,r_q)$ is a query-dependent weight vector | $\{\texttt{Sum},\texttt{Mean},\texttt{Max},\texttt{PNA}\}_{e_o \in \mathcal{N}(e_s)} \boldsymbol{m}_{(e_s,r,e_o)}^{(\ell)}$ |
| RED-GNN (Zhang & Yao, 2022) | $\alpha_{(e_s,r,e_o)\mid r_q}^{(\ell)}(\boldsymbol{h}_{e_s}^{(\ell-1)} + \boldsymbol{h}_r^{(\ell)})$, where $\alpha_{(e_s,r,e_o)\mid r_q}^{(\ell)}$ is the attention weight | $\sum_{e_o \in \mathcal{N}(e_s)} \boldsymbol{m}_{(e_s,r,e_o)}^{(\ell)}$ |

**Design space of the predictor.** From the model's perspective, we build the configuration space of the predictor and further utilize the advantages of existing structural models introduced in Sec. 2. Three query-dependent message functions $\texttt{MESS}(\cdot)$ are considered here, including DRUM (Sadeghian et al., 2019) (denoted as $\texttt{M}_\texttt{DRUM}$), NBFNet (Zhu et al., 2021) ($\texttt{M}_\texttt{NBFNet}$), and RED-GNN (Zhang & Yao, 2022) ($\texttt{M}_\texttt{REDGNN}$). The effective message is propagated from $u_q$ to the entities in subgraph $\mathcal{G}_s$.

Generally, the message propagation can be formulated as

$$\texttt{Predictor } f_{\boldsymbol{\theta}}: \boldsymbol{h}_o^{(\ell+1)} = \texttt{DROPOUT}\Big(\texttt{ACT}\Big(\texttt{AGG}\big\{\texttt{MESS}(\boldsymbol{h}_x^{(\ell)}, \boldsymbol{h}_r^{(\ell)}, \boldsymbol{h}_o^{(\ell)}):(x,r,o)\in\mathcal{E}_s\big\}\Big)\Big).$$

Note the effective message is propagated from $u$ to the entities in subgraph $\mathcal{G}_s$. The ranges for design dimensions of the configuration space are shown below, where the upper is intra-layer design while the lower is inter-layer design.

| **DROPOUT**($\cdot$) | **ACT**($\cdot$) | **AGG**($\cdot$) | **MESS**($\cdot$) | **Dimension** |
|---|---|---|---|---|
| (0, 0.5) | Identity, Relu, Tanh | Max, Mean, Sum | $\texttt{M}_\texttt{DRUM}, \texttt{M}_\texttt{NBFNet}, \texttt{M}_\texttt{REDGNN}$ | 16, 32, 64, 128 |

| **No. layers ($L$)** | **Repre. initialization** | **Layer-wise shortcut** | **Repre. concatenation** | **READOUT**($\cdot$) |
|---|---|---|---|---|
| {4, 6, 8, 10} | Binary, Relational | True, False | True, False | Linear, Dot product |

**Head and tail prediction.** Note that predicting a missing head in KG can also be formulated as tail prediction by adding inverse relations. For example, predicting a missing head in KG $(?,q,v)$ can also be formulated to tail prediction by adding inverse relations as $(v, q_\text{inverse}, ?)$. This formulation involves augmenting the original KG with inverse relations, following the approach adopted in other KG methods (Zhu et al., 2021; Zhang & Yao, 2022; Zhang et al., 2023c; Zhu et al., 2023; Zhang et al., 2023b; Galkin et al., 2023).

# C    FULL EVALUATIONS

## C.1    SETUP

**Datasets.**    We use five benchmarks with more than ten-thousand entities, including WN18RR (Dettmers et al., 2017), NELL-995 (Xiong et al., 2017), YAGO3-10 (Suchanek et al., 2007), OGBL-BIOKG and OGBL-WIKIKG2 (Hu et al., 2020). The statistics are summarized in Tab. 11.

Table 11: Statistics of the five KG datasets with more than ten-thousand entities. Fact triplets in $\mathcal{E}$ are used to build the graph, and $\mathcal{E}^{\text{train}}$, $\mathcal{E}^{\text{val}}$, $\mathcal{E}^{\text{test}}$ are edge sets of training, validation, and test set.

| dataset | $|\mathcal{V}|$ | $|\mathcal{R}|$ | $|\mathcal{E}|$ | $|\mathcal{E}^{\text{train}}|$ | $|\mathcal{E}^{\text{val}}|$ | $|\mathcal{E}^{\text{test}}|$ |
|---|---|---|---|---|---|---|
| WN18RR | 40.9k | 11 | 65.1k | 59.0k | 3.0k | 3.1k |
| NELL-995 | 74.5k | 200 | 112.2k | 108.9k | 0.5k | 2.8k |
| YAGO3-10 | 123.1k | 37 | 1089.0k | 1079.0k | 5.0k | 5.0k |
| OGBL-BIOKG | 93.7k | 51 | 5088.4k | 4762.7k | 162.9k | 162.9k |
| OGBL-WIKIKG2 | 2500.6k | 535 | 17137.1k | 16109.2k | 429.5k | 598.5k |

**Baselines.** We compare our method with general link prediction methods, including (i) semantics models: ConvE (Dettmers et al., 2017), QuatE (Zhang et al., 2019), and RotatE (Sun et al., 2019); and (ii) structural models: MINERVA (Das et al., 2017), DRUM (Sadeghian et al., 2019), RNNLogic (Qu et al., 2021), CompGCN (Vashishth et al., 2019), DPMPN (Xu et al., 2019), NBFNet (Zhu et al., 2021), and RED-GNN (Zhang & Yao, 2022). The results of these baseline are taken from their papers or reproduced by their official codes.

## C.2    COMPARISON WITH OTHER EFFICIENT METHODS

To reduce entity vocabulary to be much smaller than full entities, NodePiece (Galkin et al., 2022) represents an anchor-based approach that facilitates the acquisition of a fixed-size entity vocabulary. Note that NodePiece was originally designed for semantic models, which diverges from the primary focus of our investigation centered around structural models.

As shown below, our one-shot-subgraph link prediction method outperforms semantic models, NodePiece, and original RED-GNN in the MRR metric. Additionally, it's worth noting that NodePiece can lead to substantial performance degradation when tasked with reducing unique embeddings for a sizable number of entities. The table below illustrates this phenomenon. In contrast, the RED-GNN, when augmented with our one-shot-subgraph method, showcases an ability to enhance performance even while learning and predicting with only 10% of entities. Furthermore, the parameter count for our structural models remains significantly lower than that of semantic models, even with the incorporation of NodePiece improvements.

Table 12: Comparison with NodePiece.

|  | WN18RR | | | YAGO3-10 | | |
|---|---|---|---|---|---|---|
|  | MRR | H@10 | #Params | MRR | H@10 | #Params |
| RotatE (100% entities) | 0.476 | 57.1 | 41M | 0.495 | 67.0 | 123M |
| RotatE + NodePiece (10% entities) | 0.403 | 51.5 | 4.4M | 0.247 | 48.8 | 4.1M |
| RED-GNN (100% entities) | 0.533 | 62.4 | 0.02M | 0.559 | 68.9 | 0.06M |
| **RED-GNN + one-shot-subgraph (10% entities)** | **0.567** | **66.6** | **0.03M** | **0.606** | **72.1** | **0.09M** |

Besides, the other efficient link prediction method, DPMPN (Xu et al., 2019), contains two GNNs, one is a full-graph GNN that is similar to CompGCN, and the other one is pruned. DPMPN is a mixture of GNN methods, where the pruned GNN already knows the global information and adopts a layer-wise sampling manner. DPMPN requires several propagation steps to prune the message passing and sample the subgraph, while our PPR sampler can efficiently extract the subgraph without learning. In comparison, our method is simpler but much more effective that it observably outperforms DPMPN. Hence, considering the differences, one-shot-subgraph link prediction is still a novel subgraph sampling-based method for link prediction and achieves state-of-the-art prediction performance.

As for other sampling-based methods, the extracted subgraph by Yasunaga et al. (2021) is equivalent to the Breadth-first-searching (BFS) that is compared in our work. As it comprises all entities on the k-hop neighbors in its subgraph, the number of sampled entities could be quite large. For example,

on the WN18RR dataset, the full 5-hop neighbors of the query entity take up $9.4\%$ of entities and $13.5\%$ of edges, while the full 8-hop neighbors can even take up $69.8\%$ of entities and $89.7\%$ of edges. As for the YAGO3-10 dataset, the full 5-hop neighbors take up $98.1\%$ of entities and $95.6\%$ of edges, almost equal to the entire KG. It could still be expensive for prediction and thus not suitable for prediction on large KGs like YAGO3-10.

The sampling method of Mohamed et al. (2023) aims to extract an enclosing subgraph to answer one given query $(u, q, ?)$, which is consistent with the previous work GraIL (Teru et al., 2020). Note the extracted subgraphs are different for different given triples. That is, for answering one query $(u, q, ?)$, this method requires sampling subgraphs and predicting the score of $N$ potential links on each individual subgraph, where $N$ is the number of all entities. Hence, this method is extremely expensive and also not suitable for large KGs. By contrast, our one-shot-subgraph link prediction method only requires extracting one subgraph to answer one query rather than $N$ subgraphs. Meanwhile, the extracted subgraph is much smaller than the full k-hop neighbor. Thus, our method is more efficient and more suitable for learning and prediction on large KGs.

## C.3 THE EDGE SPLIT SCHEME

The edge split can be seen as a masking operation on KG, *i.e.*, removing the query edges from the observation edges (the inputs of a prediction model). It is necessary for KG learning; otherwise, the query edges for training can be found in the observation graph, which is not practical and not reasonable. We further clarify the edge split scheme in the following three folds.

**A general perspective of KG incompleteness.** Note that the KG datasets are generally and naturally incomplete, and the link prediction tasks aim to predict the missing links among entities. An ideal link prediction model should be robust to the intrinsic incompleteness of a KG, and the local evidence to be utilized for the prediction of a specific query is also generally incomplete. In this view, the edge split can be seen as a data augmentation method.

**Training details about edge split.** Edge split indeed influences the local connectivity of a KG, where a lower split ratio of fact:train can lead to a sparser observation graph for training. However, we afresh split the query and observation edges in each epoch, and thus, all the edges in the train set can be recursively used as the query edges for training. Besides, all these fact/train edges can be used in the test phase as factual observations. Hence, all the edges in the training set are recursively used in training and explicitly used in testing.

**The influence of edge split on training.** We conduct a further experiment with different split ratios and constrain the experiments using the same amount of training time. As the data shown below, the split ratios greatly impact the training time for one epoch, and a higher fact:train ratio leads to short training time. Besides, various split ratios only slightly influence the converged result; however, they greatly influence the speed of convergence. Specifically, a higher split ratio brings a faster convergence speed, especially on the YAGO3-10.

Table 13: Comparison of different split ratios.

|  | WN18RR | | | | YAGO3-10 | | | |
|---|---|---|---|---|---|---|---|---|
|  | MRR | H@1 | H@10 | Time | MRR | H@1 | H@10 | Time |
| Fact:Train = 0.70 | 0.566 | 51.1 | 66.7 | 22.9min | 0.552 | 46.9 | 69.9 | 55.6h |
| Fact:Train = 0.80 | 0.561 | 50.8 | 66.0 | 16.2min | 0.563 | 48.4 | 70.7 | 42.0h |
| Fact:Train = 0.90 | 0.562 | 51.1 | 66.1 | 8.7min | 0.586 | 51.4 | 71.8 | 23.2h |
| Fact:Train = 0.95 | 0.567 | 51.4 | 66.6 | 4.5min | 0.587 | 51.7 | 71.1 | 12.1h |
| Fact:Train = 0.99 | 0.563 | 51.2 | 66.3 | 0.9min | 0.598 | 52.9 | 71.9 | 2.5h |

Table 14: Comparison of effectiveness with regard to subgraph sampling.

| $r_{\mathcal{V}}^q$ | $r_{\mathcal{E}}^q$ | WN18RR | | | NELL-995 | | | YAGO3-10 | | |
|---|---|---|---|---|---|---|---|---|---|---|
|  |  | MRR | H@1 | H@10 | MRR | H@1 | H@10 | MRR | H@1 | H@10 |
| 1.0 | 1.0 | 0.549 | 50.2 | 63.5 | 0.507 | 43.8 | 61.9 | 0.598 | 53.3 | 71.2 |
| 0.5 | 0.5 | 0.555 | 50.6 | 64.4 | 0.537 | 47.3 | 64.2 | 0.599 | 53.4 | 71.4 |
| 0.2 | 0.2 | 0.563 | 51.0 | 66.0 | 0.541 | 47.9 | 63.9 | 0.603 | 53.8 | 71.8 |
| 0.1 | 0.1 | 0.567 | 51.4 | 66.4 | 0.539 | 48.0 | 63.0 | 0.599 | 53.6 | 70.8 |

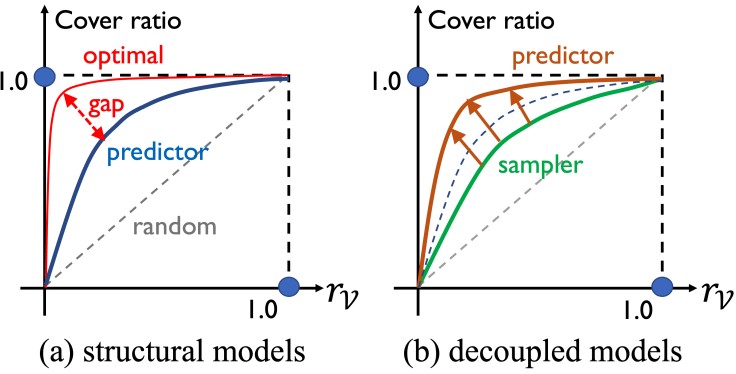

Figure 8: Illustrations of structural models (a) and decoupled subgraph models (b).

### C.4 ROBUSTNESS AGAINST DATA PERTURBATIONS

**Randomly adding noise.** We generate a noisy dataset by randomly adding noisy edges that are not observed from the dataset. Specifically, the entity set and relation set are kept the same. The noise ratio is computed as $\epsilon = |\mathcal{E}^{\text{noise}}|/|\mathcal{E}^{\text{train}}|$. The $\mathcal{E}^{\text{val}}$ and $\mathcal{E}^{\text{test}}$ are clean to guarantee the accurate evaluation.

As the results shown in Tab. 15, such noise does not significantly influence the prediction performance. Even in the high-noise scenario ($\epsilon = 50\%$), the MRR result is nearly identical to that on the clean dataset. This result shows the preliminary robustness of adding noise, and a further investigation should be conducted with more complex noise patterns, e.g., by adopting adversarial techniques.

Table 15: Model performance (MRR) with randomly added noise.

| Dataset | $\epsilon = 0\%$ (clean) | $\epsilon = 10\%$ | $\epsilon = 20\%$ | $\epsilon = 30\%$ | $\epsilon = 40\%$ | $\epsilon = 50\%$ |
|---|---|---|---|---|---|---|
| WN18RR | 0.567 | 0.567 | 0.566 | 0.566 | 0.564 | 0.564 |
| NELL-995 | 0.547 | 0.547 | 0.547 | 0.547 | 0.546 | 0.547 |
| YAGO3-10 | 0.606 | 0.605 | 0.603 | 0.601 | 0.602 | 0.601 |

**Randomly deleting facts.** We create a more incomplete KG by randomly deleting the facts in the training set. The delete ratio is computed as $r = |\mathcal{E}^{\text{delete}}|/|\mathcal{E}^{\text{train}}|$. The $\mathcal{E}^{\text{val}}$ and $\mathcal{E}^{\text{test}}$ are kept the same as the original dataset. As shown in Tab. 16, this kind of input perturbation greatly degenerates the performance, as the link prediction could rely heavily on the correct edges in the original dataset. Here, a sparser graph with more deleted edges can not sufficiently support the prediction, and the message propagation from the query entity is also hindered. Hence, combating graph incompleteness, data heterogeneity (Tang et al., 2022b;a), noisy annotations (Zhou et al., 2023a), or adversarial attacks (Zhang et al., 2022a; Chen et al., 2022; Zhou et al., 2023b; Zhang et al., 2023a; Li et al., 2023), would be a valuable direction.

Table 16: Model performance (MRR) with randomly added noise.

| Dataset | $r = 0\%$ (full) | $r = 10\%$ | $r = 20\%$ | $r = 30\%$ | $r = 40\%$ | $r = 50\%$ |
|---|---|---|---|---|---|---|
| WN18RR | 0.567 | 0.507 | 0.451 | 0.394 | 0.338 | 0.278 |
| NELL-995 | 0.547 | 0.515 | 0.483 | 0.446 | 0.412 | 0.365 |
| YAGO3-10 | 0.606 | 0.553 | 0.489 | 0.434 | 0.385 | 0.329 |

## D   FURTHER DISCUSSION

### D.1   DESIGN PRINCIPLES

Note that it is non-trivial to achieve the one-shot-subgraph link prediction, where three-fold questions are required to be answered: **(i)** From the *data*'s perspective, what kind of sampler is suitable here? **(ii)** from the *model*'s perspective, how to build up the predictor's architecture to be expressive on subgraphs? **(iii)** from the *optimization*'s perspective, how to optimize the sampler and predictor jointly and efficiently?

The major technical challenge lies in the design and implementation of an efficient and query-dependent sampler. Given a query $(u, q, ?)$, the sampler should not only preserve the local neighbors of $u$ that contain the potential answers and supporting facts, but also be able to distinguish different $q$ as the essential information could be different. Accordingly, the two design principles are as follows.

**Principle-1: Local-structure-preserving.** Since the target entity $v$ is unknown, freely sampling from all the entities in original $\mathcal{G}$ will inevitably discard the structural connection between $u$ and promising $v$. This manner of existing node-wise or layer-wise sampling methods can deteriorate the message flow started from $u$, and thus degenerate the whole prediction process. Here, the local structure of $u$ is expected to be preserved in $\mathcal{G}_s$, and each sampled entity should be reachable from $u$.

**Principle-2: Query-relation-aware.** The objective of learning relation-aware sampling here is to sample the promising targets $v$ according to the query relation $q$. Note that, in common homogeneous graphs, the relation may be unnecessary, as all relations between nodes share the same semantics. However, the relation here does matter, as it presents the distinct characteristics of a KG. Thus, in our design, the sampler is expected to be query-relation-aware in order to adapt to KGs efficiently.

## D.2 CONTRIBUTIONS

Here, we further explain the novelty and contributions in the following three folds.

**A valuable research problem.** Our investigation delves into the limitations of structural models, which confront a pronounced scalability challenge. These models rely on prediction over the entire Knowledge Graph, encompassing all entities and edges, with the additional burden of scoring all entities as potential answers. This approach proves to be highly inefficient, impeding their optimization when applied to large-scale KGs. As a consequence, we present an open question, pondering how to conduct prediction on knowledge graphs efficiently and effectively, seeking avenues to overcome this hindrance.

**A conceptual framework with several practical instantiations.** In response to the prevalent scalability challenges faced by existing KG methods, we present a conceptual solution, *i.e.*, the one-shot-subgraph link prediction manner. This novel approach promises enhanced flexibility and efficacy in design. Specifically, instead of conducting direct prediction on the complete original KG, we advocate a two-step prediction process involving subgraph sampling and subgraph-based prediction. In this regard, our proposed prediction manner comprises a sampler and a predictor. Leveraging the efficiency gains derived from subgraph-based prediction, we introduce an optimization technique for subgraph-based searching, incorporating several well-crafted technical designs. These innovations aim to strike a balance between prediction efficiency and the complexity associated with identifying optimal configurations within both data and model spaces.

**Several important discoveries from experiments.** Through comprehensive experimentation on three prevalent KGs, we demonstrate that our framework achieves state-of-the-art performance. Particularly noteworthy are the substantial advancements we achieve in both efficiency and effectiveness, a trend that is particularly evident in the case of the large-scale dataset YAGO3-10. Our quest for optimal configurations leads us to the intriguing revelation that utilizing the entire KG for prediction is unnecessary. Instead, simple heuristics can efficiently identify a small proportion of entities and facts essential for answering specific queries without the need for additional learning. These compelling findings hold significant meaning and are poised to pique the interest of the KG community.

## D.3 EXPLANATION FOR THE IMPROVEMENT IN PERFORMANCE

Here, we provide a two-fold explanation for better performance as follows.

**Data perspective:** Extracting subgraphs can remove irrelevant information for link prediction. Conventional structural models explicitly take all the entities and edges into prediction, ignoring the correlation between the entities and the query relation. As delineated in Sec. 5 of this paper, our one-shot-subgraph link prediction approach effectively discerns and excludes irrelevant entities while retaining the proper answers. This strategic refinement, in turn, contributes to simplifying the learning problem, thereby amplifying prediction performance. Our findings demonstrate that a mere fraction (*i.e.*, 10%) of entities suffices for answering specific queries, *i.e.*, a subset efficiently identified by the heuristic Personalized PageRank mechanism without the need for learnable sampling.

Another supporting material is that only relying on a subgraph for prediction can also boost the test-time performance (Miao et al., 2022). The proposed GAST method (Miao et al., 2022), aims to extract a subgraph $G_s$ as the interpretation of a GNN. It inherits the same spirit of information bottleneck in building its optimization objective, *i.e.*, $\min -I(G_s; Y) + \beta \cdot I(G_s; G)$. The integrated subgraph sampler can explicitly remove the spurious correlation or noisy information in the entire graph $G$, which is similar to our one-shot-subgraph link prediction framework.

**Model perspective:** The higher learning efficiency on subgraphs can further boost hyper-parameter optimization (Zhang et al., 2022b). Generally, the task of hyperparameter optimization on extensive graph datasets poses substantial challenges due to the inherent inefficiencies in model training. However, the implementation of our subgraph-based method results in a substantial improvement in training efficiency.

Consequently, we can rapidly obtain evaluation feedback for configurations sampled from the hyperparameter space. Sec. 4.2 introduces an optimization technique for subgraph-based searching that features meticulously crafted technical designs. The searched configuration usually leads to a deeper GNN (*i.e.*, 8 layers, while previous studies are usually limited to 5 or 6 layers) that increases the expressiveness of prediction. These innovations aim to balance prediction efficiency and the complexity associated with identifying optimal configurations within data and model spaces, which also contributes to improved prediction performance.

### D.4 EXTENSION

Note that KG learning usually focuses more on link-level tasks, *e.g.*, link prediction. However, we firmly believe that one-shot-subgraph link prediction has the potential to be extended and adapted for various graph learning tasks. One general direction is to adapt the one-shot-subgraph link prediction framework to other kinds of graph learning tasks, *e.g.*, the node-level or graph-level tasks.

For instance, with the PPR sampler, one can sample a single-source "local" subgraph for node classification or a multi-source "global" subgraph for graph classification, where the rationale of first sampling and then prediction remains applicable. Besides, enhancing the one-shot-subgraph link prediction with instance-wise adaptation is also a promising direction. That is, sampling a subgraph of suitable scale for each given query, which can potentially improve the upper limit of prediction.

Furthermore, conducting link prediction with new relations or new entities is also a frontier topic. Improving the generalization or extrapolation power of GNN can be vital in practice. Considering the significant few-shot in-context learning of the large language model (LLM), an appropriate synergy between the latest LLM and current GNN will be a promising direction. Improving the efficiency and scalability of predicting with large graphs is also of great importance here.

In addition, from a broader perspective of trustworthy machine learning, one should also consider the intrinsic interpretability and the robustness problem. These trustworthy properties can help users understand the model better and also keep it in a safe and controllable way.

# E CASE STUDY

In this section, we show the sampled subgraph of different scales on three datasets as follows.

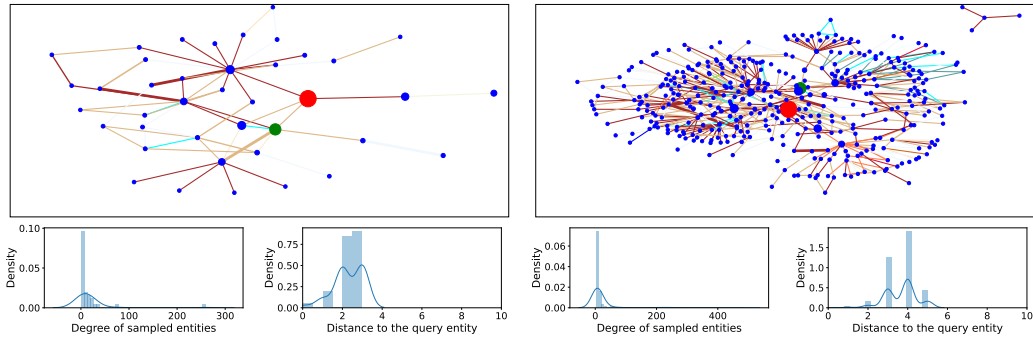

Figure 9: Subgraphs (0.1% and 1%) from WN18RR: $u=1, q=12, v=\{5305\}$.

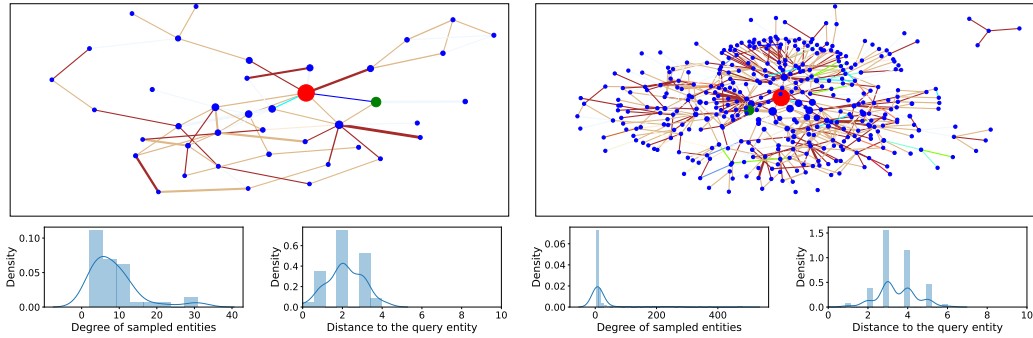

Figure 10: Subgraphs (0.1% and 1%) from WN18RR: $u=9, q=20, v=\{38116\}$.

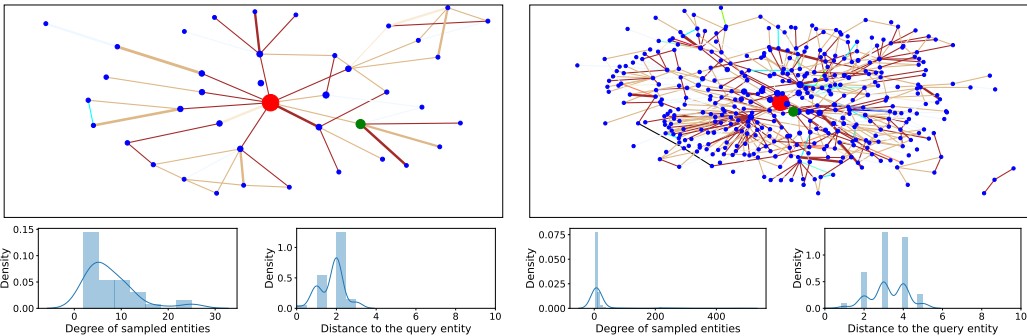

Figure 11: Subgraphs (0.1% and 1%) from WN18RR: $u=29, q=1, v=\{11186\}$.

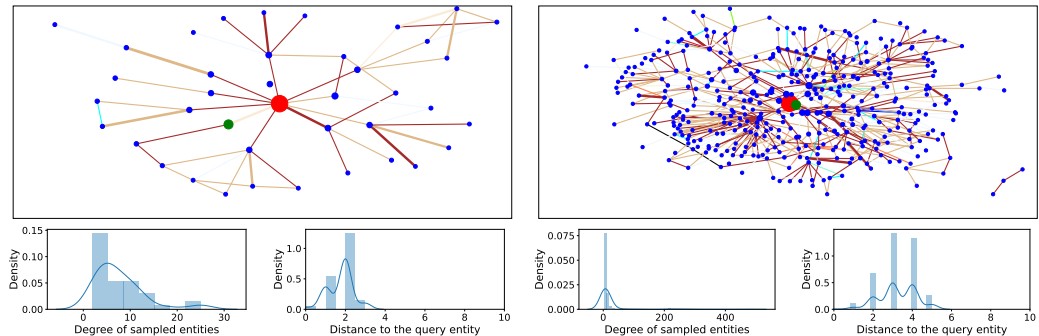

Figure 12: Subgraphs (0.1% and 1%) from WN18RR: $u=29, q=12, v=\{6226\}$.

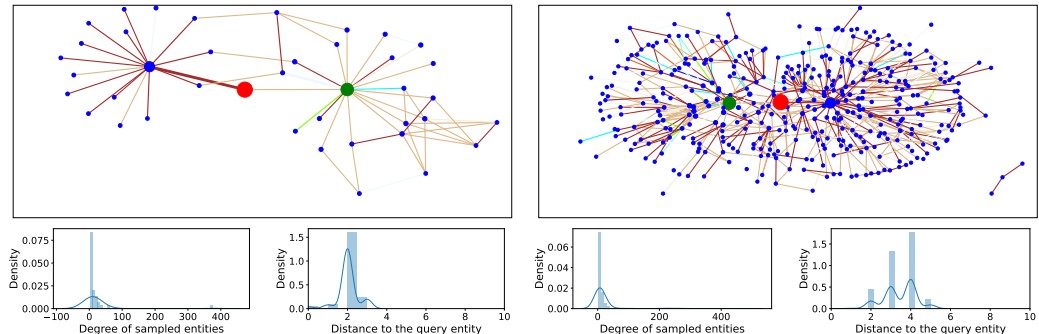

Figure 13: Subgraphs (0.1% and 1%) from WN18RR: $u=44, q=12, v=\{45\}$.

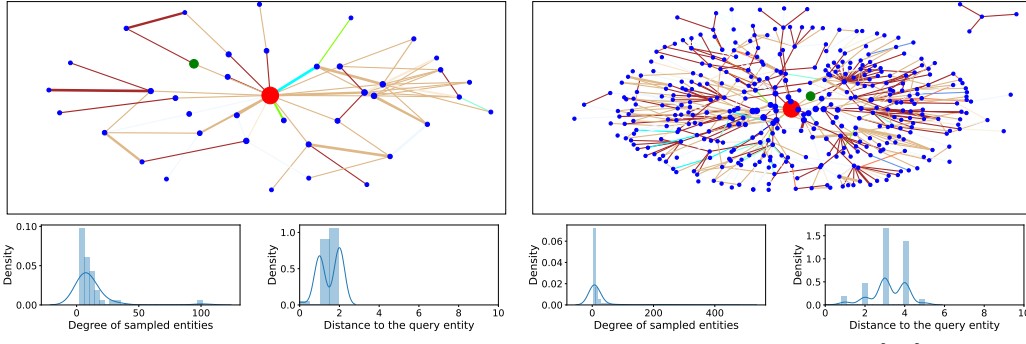

Figure 14: Subgraphs (0.1% and 1%) from WN18RR: $u=45, q=1, v=\{44\}$.

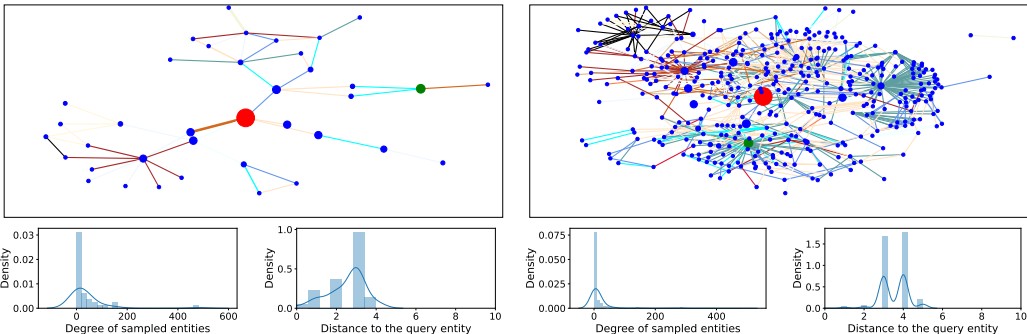

Figure 15: Subgraphs (0.1% and 1%) from WN18RR: $u=60, q=2, v=\{6577\}$.

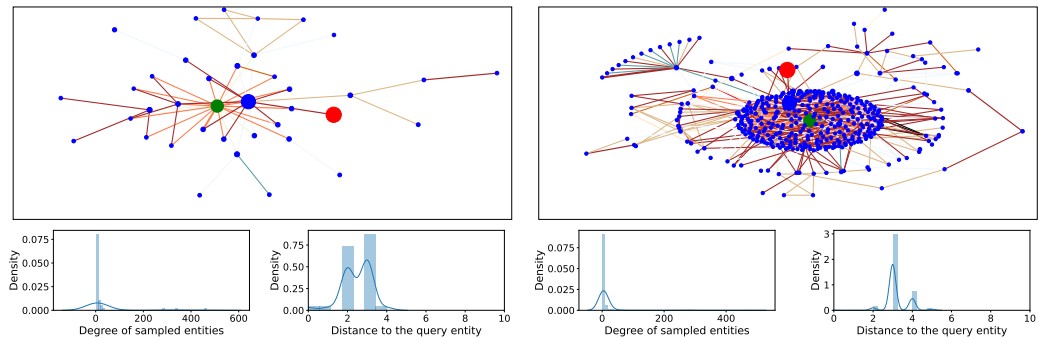

Figure 16: Subgraphs (0.1% and 1%) from WN18RR: $u=78, q=5, v=\{172\}$.

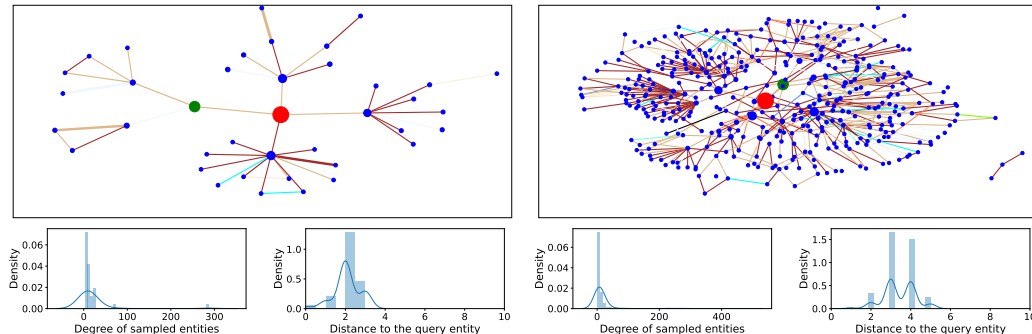

Figure 17: Subgraphs (0.1% and 1%) from WN18RR: $u=88, q=12, v=\{4621\}$.

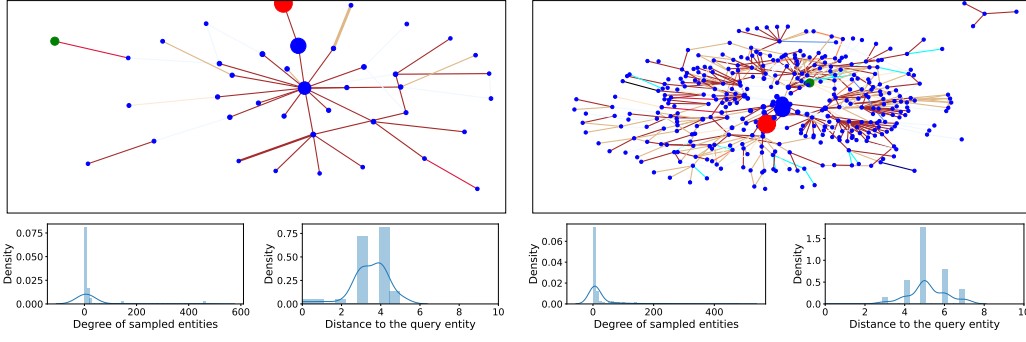

Figure 18: Subgraphs (0.1% and 1%) from WN18RR: $u=155, q=19, v=\{785\}$.

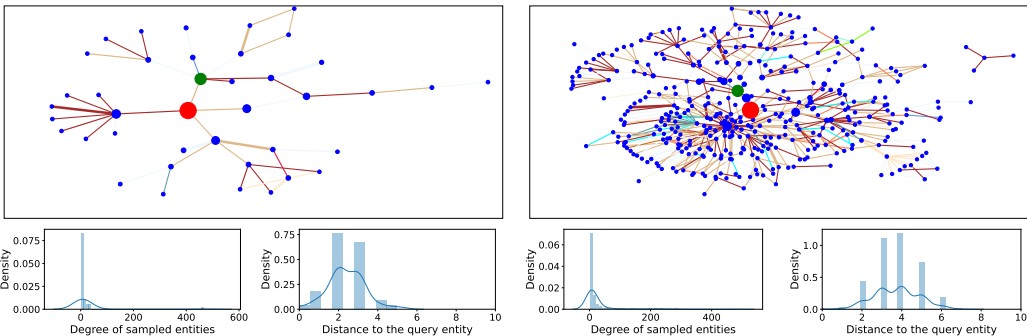

Figure 19: Subgraphs (0.1% and 1%) from WN18RR: $u=3297, q=1, v=\{2037\}$.

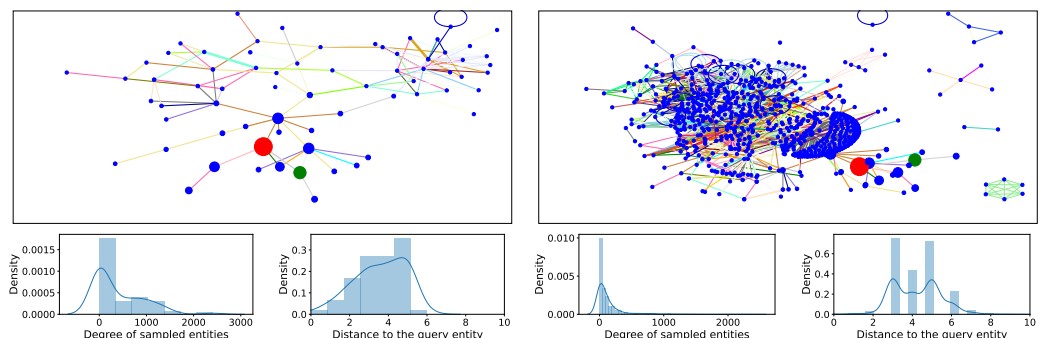

Figure 20: Subgraphs (0.1% and 1%) from NELL-995: $u = 4, q = 238, v = \{22677\}$.

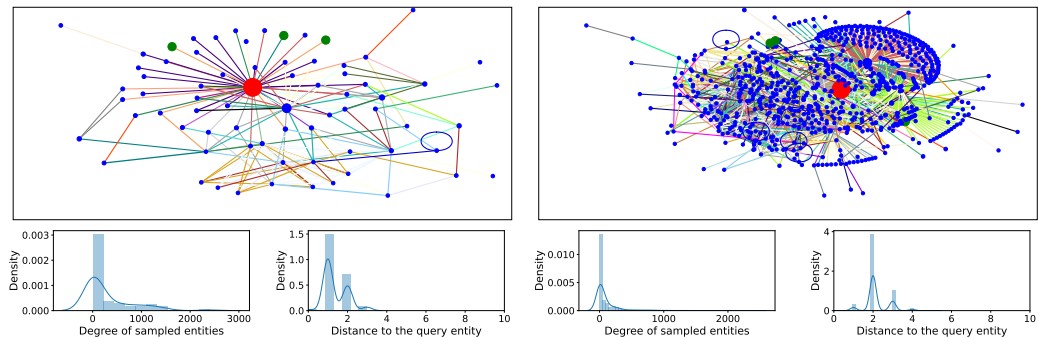

Figure 21: Subgraphs (0.1% and 1%) from NELL-995: $u = 17, q = 274, v = \{57735, 61381, 63044\}$.

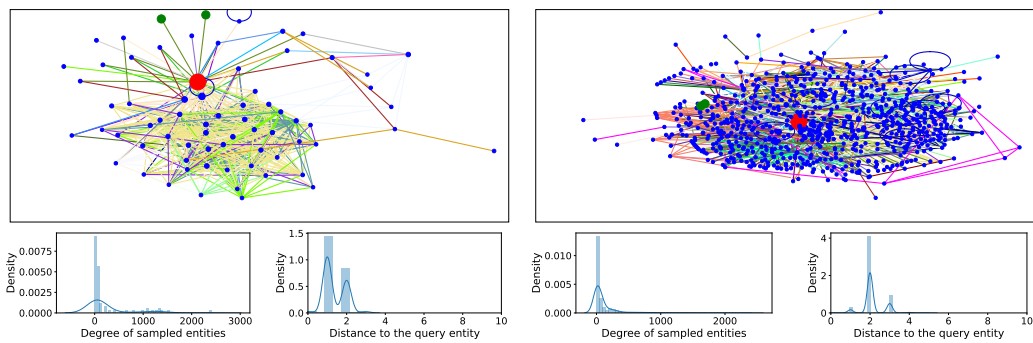

Figure 22: Subgraphs (0.1% and 1%) from NELL-995: $u = 29, q = 260, v = \{27725, 73985\}$.

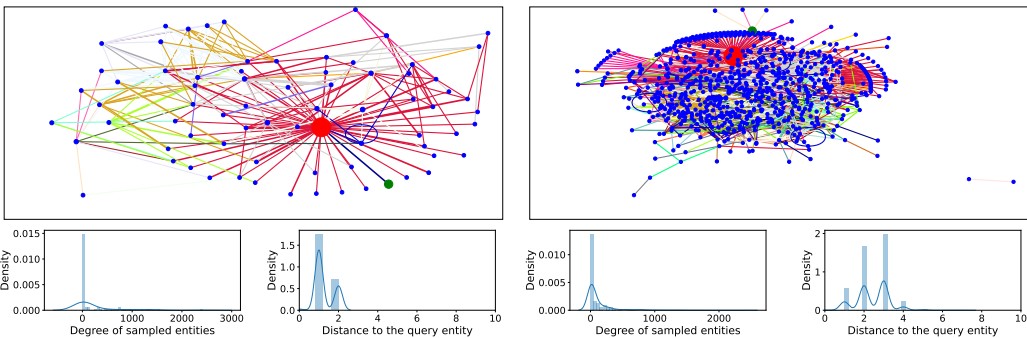

Figure 23: Subgraphs (0.1% and 1%) from NELL-995: $u = 44, q = 222, v = \{11669\}$.

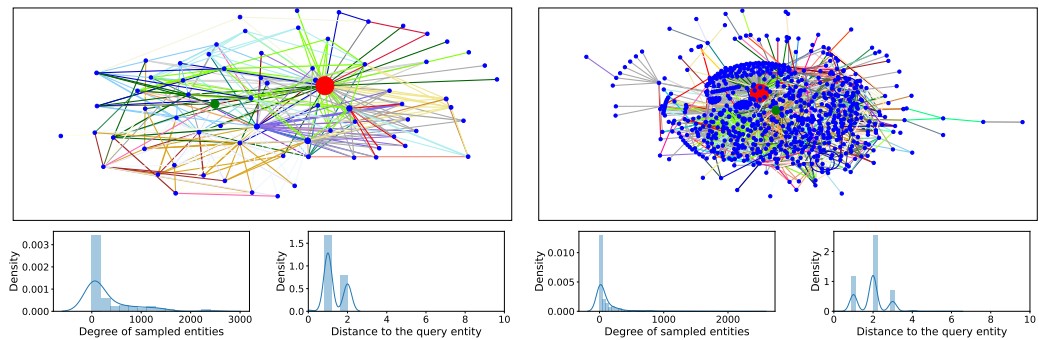

Figure 24: Subgraphs (0.1% and 1%) from NELL-995: $u = 60, q = 74, v = \{164\}$.

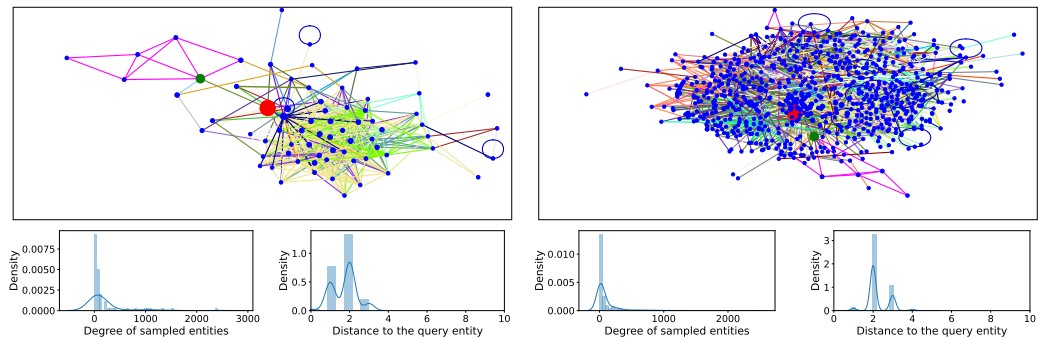

Figure 25: Subgraphs (0.1% and 1%) from NELL-995: $u = 166, q = 260, v = \{6364\}$.

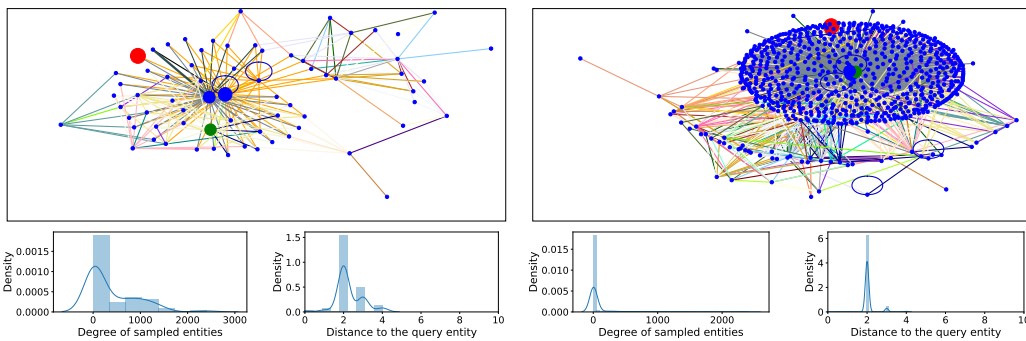

Figure 26: Subgraphs (0.1% and 1%) from NELL-995: $u = 202, q = 101, v = \{399\}$.

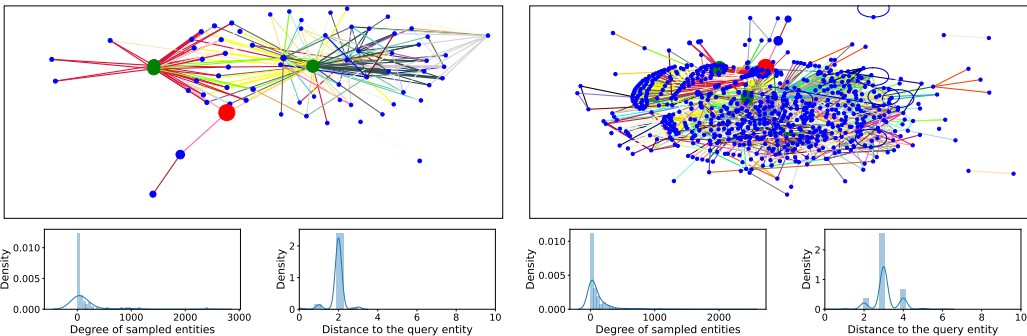

Figure 27: Subgraphs (0.1% and 1%) from NELL-995: $u = 255, q = 232, v = \{1631, 9925, 11229\}$.

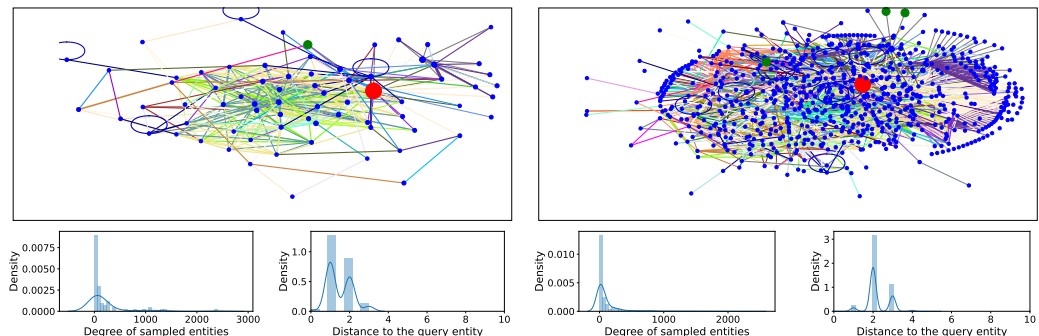

Figure 28: Subgraphs (0.1% and 1%) from NELL-995: $u{=}1371, q{=}260, v{=}\{24193, 50385, 60718\}$.

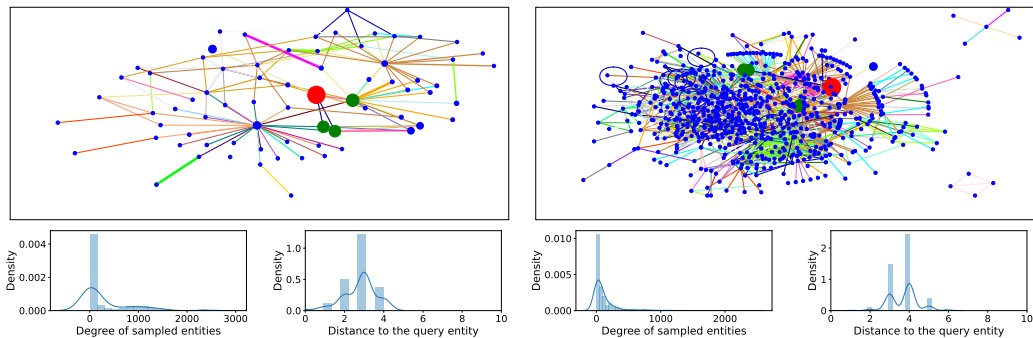

Figure 29: Subgraphs (0.1% and 1%) from NELL-995: $u{=}11200, q{=}38, v{=}\{5737, 7292, 11199\}$.

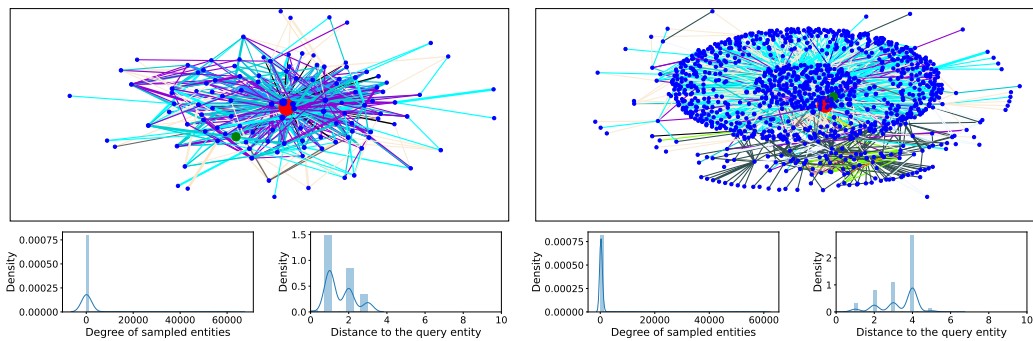

Figure 30: Subgraphs (0.1% and 1%) from YAGO3-10: $u{=}17, q{=}39, v{=}\{54968\}$.

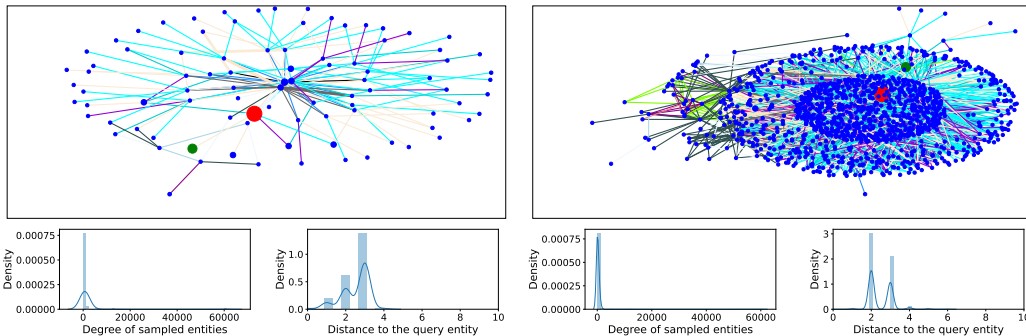

Figure 31: Subgraphs (0.1% and 1%) from YAGO3-10: $u{=}20, q{=}2, v{=}\{34580\}$.

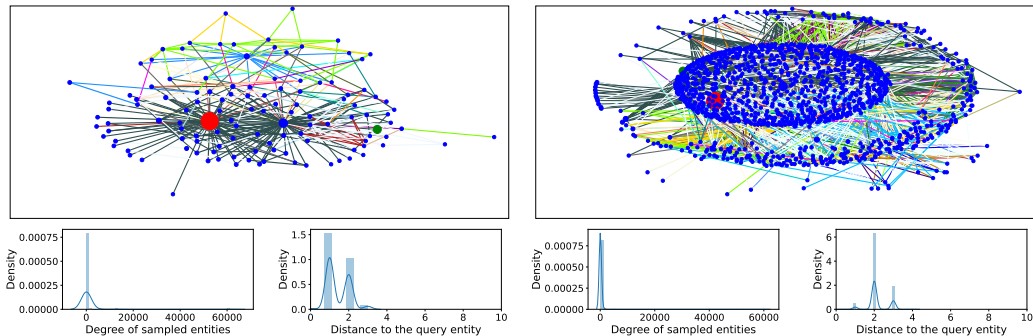

Figure 32: Subgraphs (0.1% and 1%) from YAGO3-10: $u\!=\!25, q\!=\!37, v\!=\!\{40490\}$.

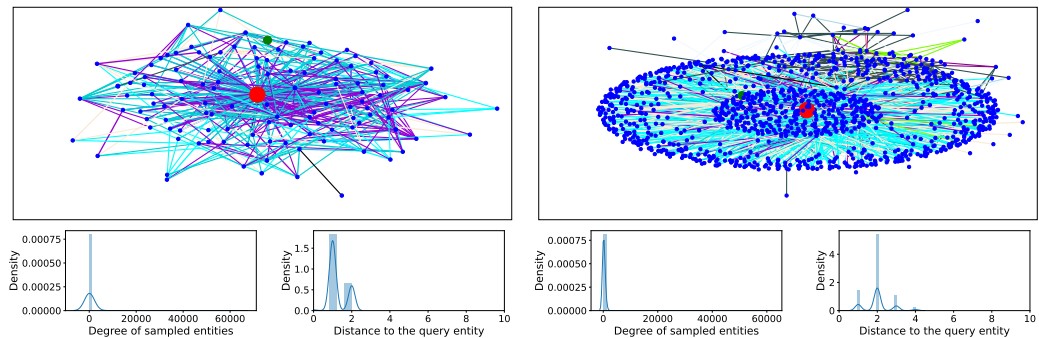

Figure 33: Subgraphs (0.1% and 1%) from YAGO3-10: $u\!=\!27, q\!=\!39, v\!=\!\{3801\}$.

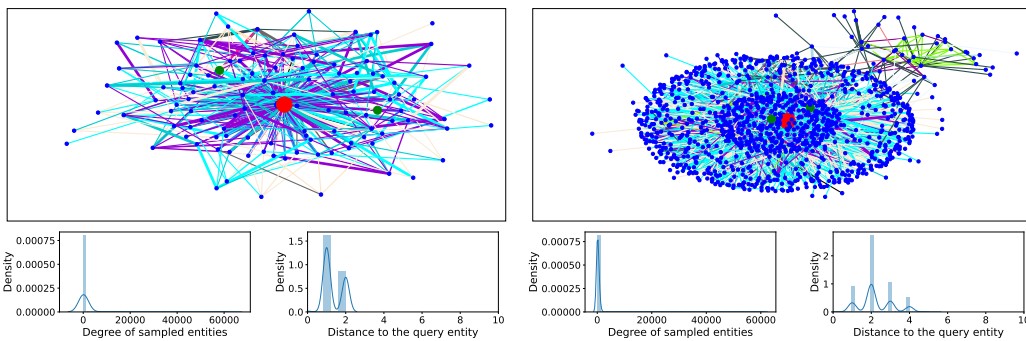

Figure 34: Subgraphs (0.1% and 1%) from YAGO3-10: $u\!=\!29, q\!=\!38, v\!=\!\{33723, 82573\}$.

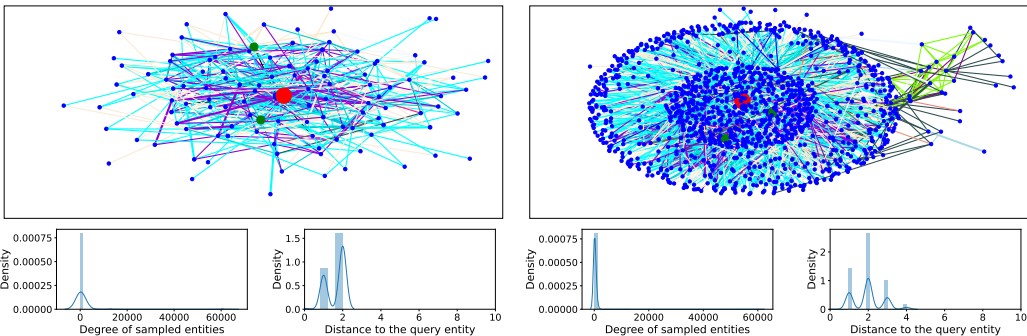

Figure 35: Subgraphs (0.1% and 1%) from YAGO3-10: $u\!=\!55, q\!=\!38, v\!=\!\{14834, 67740\}$.

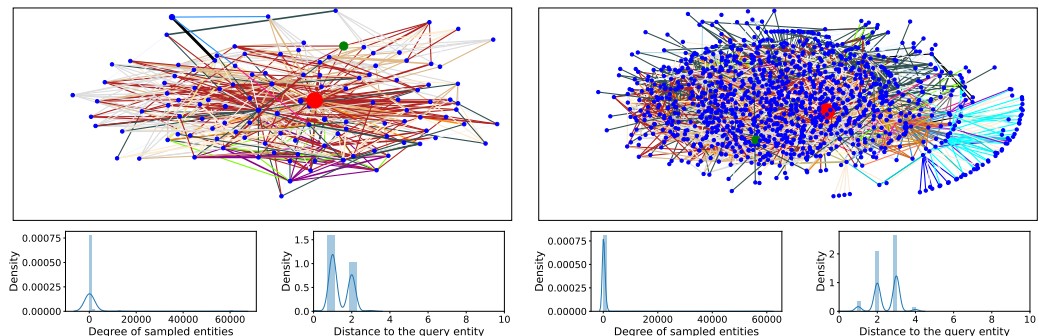

Figure 36: Subgraphs (0.1% and 1%) from YAGO3-10: $u=102, q=12, v=\{15823\}$.

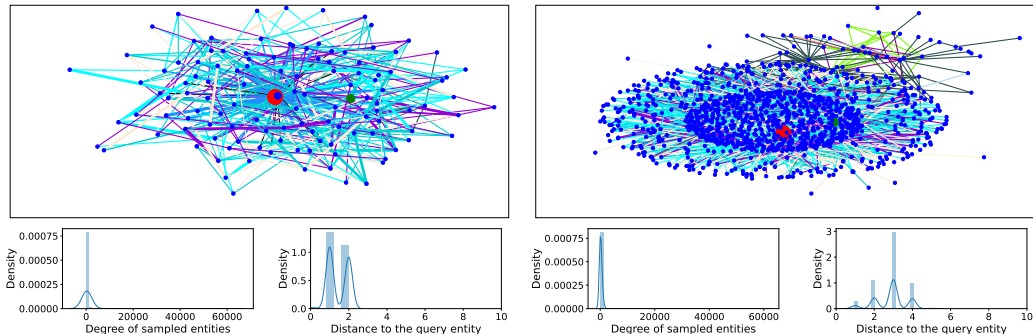

Figure 37: Subgraphs (0.1% and 1%) from YAGO3-10: $u=108, q=39, v=\{7271\}$.

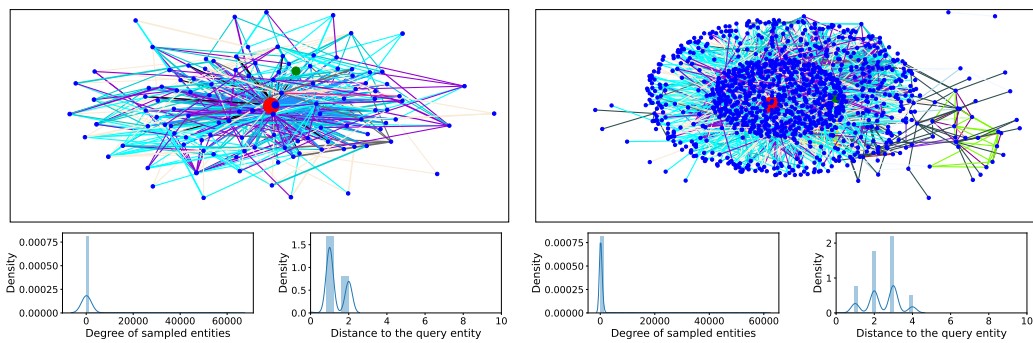

Figure 38: Subgraphs (0.1% and 1%) from YAGO3-10: $u=135, q=38, v=\{51096\}$.

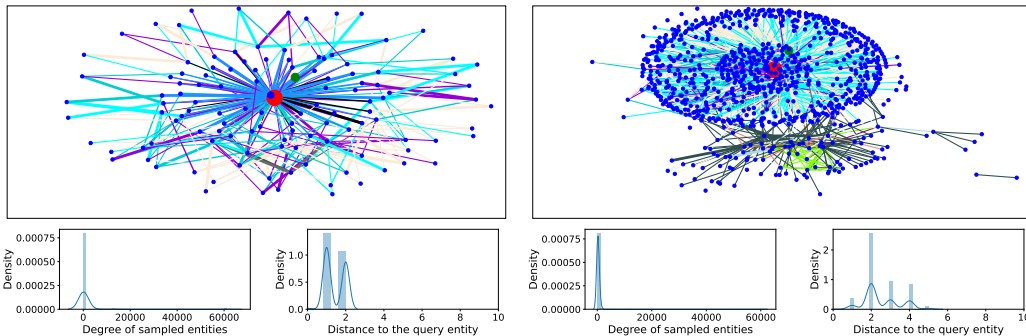

Figure 39: Subgraphs (0.1% and 1%) from YAGO3-10: $u=137, q=39, v=\{26722\}$.

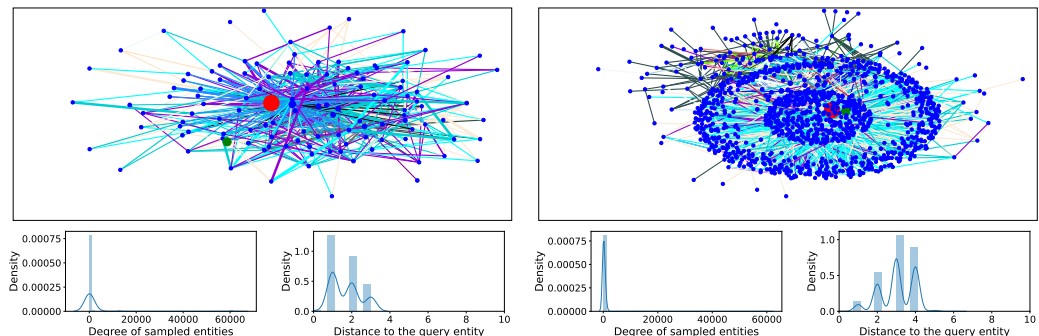

Figure 40: Subgraphs (0.1% and 1%) from YAGO3-10: $u\!=\!446, q\!=\!38, v\!=\!\{104297\}$.

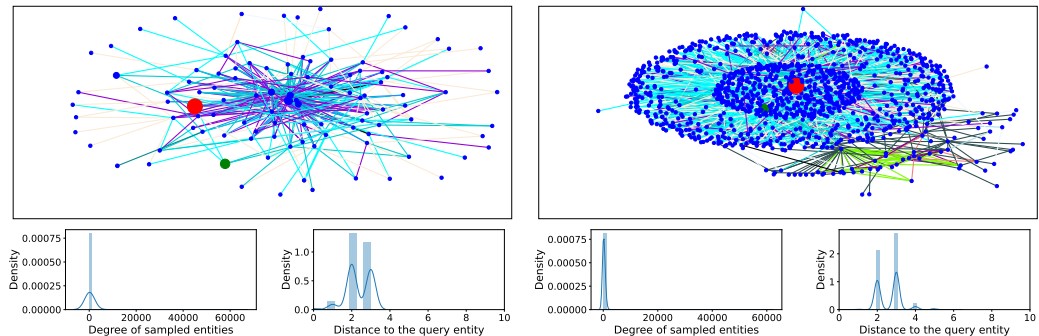

Figure 41: Subgraphs (0.1% and 1%) from YAGO3-10: $u\!=\!1072, q\!=\!1, v\!=\!\{23394\}$.

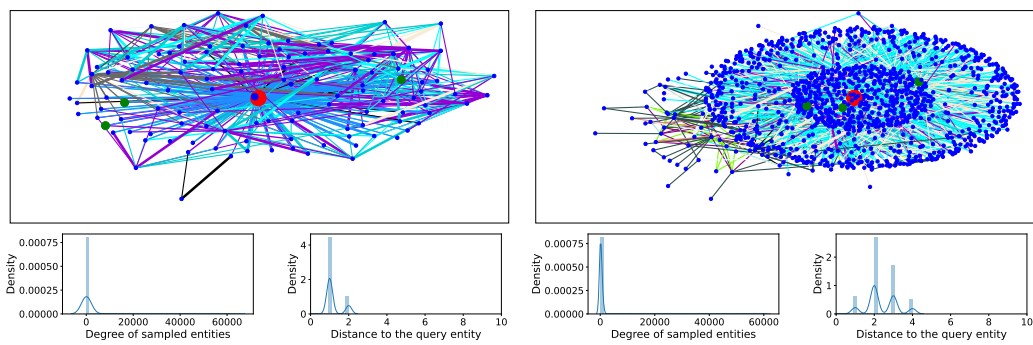

Figure 42: Subgraphs (0.1% and 1%) from YAGO3-10: $u\!=\!1255, q\!=\!38, v\!=\!\{12418, 28138, 71366\}$.

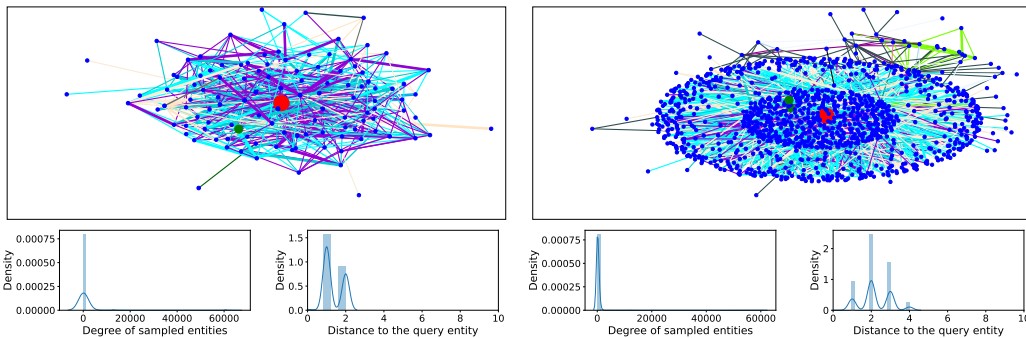

Figure 43: Subgraphs (0.1% and 1%) from YAGO3-10: $u\!=\!2252, q\!=\!39, v\!=\!\{9476, 77502\}$.

