# OpenReview forum: "Less is More: One-shot Subgraph Reasoning on Large-scale Knowledge Graphs"
_ICLR.cc/2024/Conference — ICLR 2024 poster_

### Official Review · Reviewer_FPeS · 2023-10-19

**Soundness:** 3 good
**Presentation:** 3 good
**Contribution:** 3 good
**Rating:** 5
**Confidence:** 4

**Summary:**

This paper aims to address the serious scalability problem of using the entire knowledge graph (KG) for inference. To this end, one-shot subgraph reasoning on large-scale KG is proposed. This method decouples the inference process into two steps, (i) extracting only one query-dependent subgraph and (ii) reasoning on this single subgraph. This method has higher training efficiency and stronger reasoning ability.

**Strengths:**

1.The motivation of model design is clear and reasonable. It is unnecessary to utilize the whole KG in reasoning, only a small proportion of entities and facts are essential for answering specific queries, which is also supported by the experiments.

2.Experiments cover several benchmarks. The model is tested on multiple datasets and shows very promising results.

3.The paper is well-written and generally easy to follow.

**Weaknesses:**

1.In the paper, there is some similar descriptions in the discussion. For example, in section 3, the first paragraph states, 'The design principle here is to first identify one subgraph, which is relevant to a given query and is much smaller than the original graph, and then effectively reason on this subgraph to obtain the precise prediction.' This is similar to Definition 1 in section 3, which states, 'Instead of directly reasoning on the original graph G, the reasoning procedure is decoupled into two-fold: (1) one-shot sampling of a query-dependent subgraph and (2) reasoning on this subgraph.' These are also similar to what is mentioned in the introduction, 'Thereby, the reasoning of a query is conducted by (i) fast sampling of one query-dependent subgraph with the one-shot sampler and (ii) slow reasoning on this single subgraph with the predictor.'

2.In sections 4.4, a significant number of symbols, abbreviations, and technical terms are employed. Some symbols are not adequately explained, which may potentially cause difficulties for readers during their reading. For example, the supp() in 4.4Theorem 1 does not provide an explanation.

3.The formulas in sections 4.3 and 4.4 are not labeled, such as (6) and (7).

**Questions:**

1.Given that the proposed method relies on a non-parametric heuristic (PPR) for sampling, how interpretable are the final predictions and reasoning steps? Can the method provide insights into why a certain answer was chosen for a given query?

2.How robust is the proposed method to noise or inaccuracies in the underlying knowledge graph? Are there any guarantees regarding the robustness of reasoning results in the presence of perturbations in the data?

3.What are the potential directions for future research or extensions of this work? Are there specific aspects or challenges within KG reasoning that remain open for investigation?

---

> ### Author Response · Authors · 2023-11-19
> **Response to Reviewer FPeS (1/3)**
>
> We thank reviewer FPeS for the valuable feedback. We addressed all the comments you provided. Please find the point-to-point responses below. Any further comments and discussions are welcome!
>
> **Q1.** About the writing.
>
> > Q1.1 In the paper, there is some similar descriptions in the discussion.
> > Q1.2 In sections 4.4, a significant number of symbols, abbreviations, and technical terms are employed. Some symbols are not adequately explained, which may potentially cause difficulties for readers during their reading. For example, the supp() in 4.4 Theorem 1 does not provide an explanation.
> > Q1.3 The formulas in sections 4.3 and 4.4 are not labeled, such as (6) and (7).
>
> **Reply:** Thanks for your kind suggestions. **We have made several modifications to our draft. Please refer to our latest draft, where the modified contents are highlighted in blue.** Specifically,
>
> - **Word use.** We update the words used in our draft, e.g., "paradigm" -> "manner", "KG reasoning" -> "link prediction", and "reasoning" -> "prediction". The keyword of our method is fine-tuned to "one-shot subgraph link prediction" and formally defined in Def. 1. We delete those repeated claims, e.g., the two-step prediction pipeline with sampling and predicting.
> - **Technical details.** We move some technical details from the appendix to the main contents. Please refer to Sec. 4.2, where the implementation details of search space, search problem, and search algorithm are elaborated on. Besides, for a better understanding, we also add Fig. 2 to illustrate the technical details.
> - **Experimental intuitions.** The intuitions of the experiments are further explained and highlighted in Sec. 5. For example, in Fig. 5, the below distributions of degree and distance show the statistical properties of each subgraph. More importantly, the visualization of Fig. 5 shows that the local structure of query entity $u$ is well preserved, while the true answers $v$ are also covered in the subgraphs, which contributes to the model's performance.
> - **Mathematical symbols and equations.** We add more introduction to the used variables in Sec. 4.3 and leave the detailed proof in Appendix A.3. We also label the formulas in Secs. 4.2 and 4.3 with numbers.
>
> We thoroughly considered the concerns the reviewer pointed out and tried our best to improve the contents. Let us know if some content needs to be further modified, and we will modify it as soon as possible.

---

> ### Author Response · Authors · 2023-11-19
> **Response to Reviewer FPeS (2/3)**
>
> **Q2.** Interpretability and insights of the PPR sampling.
>
> > Given that the proposed method relies on a non-parametric heuristic (PPR) for sampling, how interpretable are the final predictions and reasoning steps? Can the method provide insights into why a certain answer was chosen for a given query?
>
> **Reply:** Thanks for the valuable feedback. If a certain answer is chosen for a given query by our method, it should **(1) remain in the subgraph by the sampler** and **(2) get a high score with a relatively high rank by the predictor**.
>
> Speficically, for **(1)**,
>
> - Our experiments in Tab. 3 and Fig. 3 reveal that PPR gets a much higher Coverage Ratio (CR) and notably outperforms other heuristics (BFS, PR, RW, RAND) in identifying potential answers.
> - The reason is that PPR is query-dependent and local-structure-preserving for its single-source scoring that starts from $u$. Previous studies also show that $v$ is generally near to $u$, and the relational paths connecting $u$ and $v$ that support the query also lie close to $u$.
>
> - According to the visualized subgraphs in Fig. 5, the local structure of query entity $u$ is well preserved, while the true answers $v$ are also covered in the subgraphs, which contributes to the model's performance.
>
> - Besides, the extracted subgraphs are interpretable to some extent, which shares the same manner of interpretable graph learning that aims to identify the key subgraphs for predictions [1, 2].
>
> As for **(2)**,
>
> - Our method with PPR achieves leading performances on all five large-scale benchmarks over all the baselines, especially on the largest YAGO3-10 and OGB datasets. If we replace the PPR with other heuristics, the final performance will also decrease significantly, as shown in Tab. 5.
> - Importantly, we reveal that it is unnecessary to utilize the whole KG in link prediction; meanwhile, only a small proportion of entities and facts are essential for answering specific queries that can be quickly identified by the PPR heuristics.
>
> Further, we provide a two-fold explanation regarding the better performance of the joint prediction, including the above two steps.
>
> - **Data perspective: Extracting subgraphs can remove irrelevant information for link prediction.** Conventional structural models explicitly take all the entities and edges into prediction, while our one-shot-subgraph method discerns and excludes irrelevant entities while retaining the proper answers. This strategic refinement, in turn, contributes to simplifying the learning problem, thereby amplifying prediction performance.
> - Another supporting material is that only relying on a subgraph for prediction can also boost the test-time performance [2]. The method [2] extracts a subgraph $G_s$ as the interpretation of a GNN, wherein the integrated subgraph sampler can explicitly remove the spurious correlation or noisy information in the original $G$.
> - **Model perspective: The higher learning efficiency on subgraphs can further boost hyper-parameter optimization.** Generally, hyperparameter optimization on large-scale datasets poses challenges due to the inherent inefficiencies in model training. However, the implementation of our subgraph-based method results in a substantial improvement in training efficiency.
> - Consequently, we can rapidly obtain evaluation feedback for configurations sampled from the hyperparameter space. The searched configuration usually leads to a deeper GNN (i.e., 8 layers, while previous studies are usually limited to 5 or 6 layers) that increases the expressiveness of prediction. These data-adaptive configurations also contribute to improved prediction performance.
>
> [1] GNNExplainer: Generating Explanations for Graph Neural Networks. NeurIPS 2019.
>
> [2] Interpretable and Generalizable Graph Learning via Stochastic Attention Mechanism. ICML 2022.

---

> ### Author Response · Authors · 2023-11-19
> **Response to Reviewer FPeS (3/3)**
>
> **Q3.** Robustness against data perturbations.
>
> > How robust is the proposed method to noise or inaccuracies in the underlying knowledge graph? Are there any guarantees regarding the robustness of reasoning results in the presence of perturbations in the data?
>
> **Reply:** Thanks for the insightful comment about the noise aspect. Although noise robustness is not our focus, we conduct empirical studies with two kinds of input perturbations.
>
> **Randomly adding noise**: We generate a noisy dataset by randomly adding noisy edges that are not observed from the dataset. Specifically, the entity set and relation set are kept the same. The noise ratio is computed as $\epsilon=\frac{|\mathcal{E}^{\text{noise}}|}{|\mathcal{E}^{\text{train}}|}$, and the $\mathcal{E}^{\text{val}}$ and $\mathcal{E}^{\text{test}}$ are clean to guarantee an accurate evaluation.
>
> As the results (in MRR) shown below, such noise does not significantly influence the prediction performance. Even in the high-noise scenario ($\epsilon=50$%), the MRR result is nearly identical to that on the clean dataset. This result shows the preliminary robustness of the noise, and further investigation should be conducted with more complex noise patterns, e.g., by adopting adversarial techniques.
>
> | Dataset / Noise ratio $\epsilon$ | 0% (clean) | 10%   | 20%   | 30%   | 40%   | 50%   |
> | -------------------------------- | ---------- | ----- | ----- | ----- | ----- | ----- |
> | WN18RR                           | 0.567      | 0.567 | 0.566 | 0.566 | 0.564 | 0.564 |
> | NELL-995                         | 0.547      | 0.547 | 0.547 | 0.547 | 0.546 | 0.547 |
> | YAGO3-10                         | 0.606      | 0.605 | 0.603 | 0.601 | 0.602 | 0.601 |
>
> **Randomly deleting facts**: We create a more incomplete KG by randomly deleting the facts in the training set. The delete ratio is computed as $r=\frac{|\mathcal{E}^{\text{delete}}|}{|\mathcal{E}^{\text{train}}|}$, and the $\mathcal{E}^{\text{val}}$ and $\mathcal{E}^{\text{test}}$ are kept the same as the original dataset.
>
> As shown below, this kind of input perturbation greatly degenerates the performance, as the link prediction could rely heavily on the correct edges in the original dataset. Here, a sparser graph with more deleted edges cannot sufficiently support the prediction, and the message propagation from the query entity is also hindered. Hence, combating the graph's incompleteness would be a valuable direction.
>
> | Dataset / Delete ratio $r$ | 0% (full) | 10%   | 20%   | 30%   | 40%   | 50%   |
> | -------------------------- | --------- | ----- | ----- | ----- | ----- | ----- |
> | WN18RR                     | 0.567     | 0.507 | 0.451 | 0.394 | 0.338 | 0.278 |
> | NELL-995                   | 0.547     | 0.515 | 0.483 | 0.446 | 0.412 | 0.365 |
> | YAGO3-10                   | 0.606     | 0.553 | 0.489 | 0.434 | 0.385 | 0.329 |
>
> **Q4.** Discussion on the potential directions.
>
> > What are the potential directions for future research or extensions of this work? Are there specific aspects or challenges within KG reasoning that remain open for investigation?
>
> **Reply:** Thanks for the comment. Note that KG learning usually focuses more on link-level tasks, e.g., link prediction. However, we firmly believe that one-shot-subgraph link prediction has the potential to be extended and adapted for various graph learning tasks and scenarios.
>
> One general direction is to adapt the one-shot-subgraph link prediction framework to other kinds of graph learning tasks, e.g., node-level or graph-level tasks. For instance, with the PPR sampler, one can sample a single-source "local" subgraph for node classification or a multi-source "global" subgraph for graph classification, where the rationale of first sampling and then prediction remains applicable. Besides, enhancing the one-shot-subgraph link prediction with instance-wise adaptation is also a promising direction. That is, sampling a subgraph of suitable scale for each given query, which can potentially improve the upper limit of prediction.
>
> Besides, conducting link prediction with new relations or new entities is also a frontier topic. Improving the generalization or extrapolation power of GNN can be vital in practice. Considering the significant few-shot in-context learning of the large language model (LLM), an appropriate synergy between the latest LLM and current GNN will be a promising direction. Improving the efficiency and scalability of predicting with large graphs is also of great importance here.
>
> In addition, from a broader perspective of trustworthy machine learning, one should also consider the intrinsic interpretability discussed in Q2 and the robustness problem discussed in Q3. These trustworthy properties can help users understand the model better and also keep it in a safe and controllable way.
>
> The above contents have been added to our latest draft and highlighted in blue.

---

> ### Author Response · Authors · 2023-11-21
> **Would you mind checking our responses and confirming whether you have any further questions?**
>
> Dear Reviewer FPeS,
>
> Thanks very much for your time and valuable comments.
>
> In the rebuttal period, we have provided detailed responses to all your comments and questions point-by-point for the unclear presentations. Specifically, we
> - improve the writing to be more clear and informative (**Q1**);
> - clarify the interpretability and insights of the PPR sampling (**Q2**);
> - conduct experiments to evaluate the robustness against data perturbations (**Q3**);
> - discuss the potential directions (**Q4**).
>
> Would you mind checking our responses and confirming whether you have any further questions?
>
> Any comments and discussions are welcome!
>
> Thanks for your attention and best regards.
>
> Authors of #3620

---

> ### Author Response · Authors · 2023-11-22
> **We are anticipating your post-rebuttal feedback!**
>
> Dear Reviewer FPeS,
>
> Thanks very much for your time and valuable comments.
>
> In the rebuttal period, we have provided detailed responses to all your comments and questions and made significant modifications in the revision.
>
> **We understand you might be quite busy. However, as the discussion deadline is approaching, would you mind checking our response and confirming whether you have any further questions?**
>
> Any comments and discussions are welcome!
>
> Thanks for your attention and best regards.
>
> Authors of #3620

---

> ### Author Response · Authors · 2023-11-23
> **[Last-day Reminder] We are anticipating your post-rebuttal feedback!**
>
> Dear Reviewer FPeS,
>
> Thanks very much for your time and valuable comments.
>
> We understand you might be quite busy. **However, the discussion deadline is approaching, and we have only a few hours left.**
>
> Would you mind checking our response and confirming whether you have any further questions?
>
> Thanks for your attention.
>
> Best regards,
>
> Authors of #3620

---

### Official Review · Reviewer_RRjT · 2023-10-31

**Soundness:** 3 good
**Presentation:** 3 good
**Contribution:** 3 good
**Rating:** 6
**Confidence:** 3

**Summary:**

The paper studies the reasoning on large-scale knowledge graphs and proposes to extract only one query-dependent subgraph, then reasoning on this single subgraph. Specifically, the authors propose to use personalized PageRank to assign probabilities, and then extract the subgraph according to the probabilities. Three different query-dependent message functions are used in the reasoning step. The experiments are conducted on three common datasets.

**Strengths:**

1. The paper is well-motivated. It is reasonable to consider the one-shot subgraph reasoning problem.
2. The paper is well-written and easy to follow.
3. The paper provides a comprehensive literature review in preliminaries.

**Weaknesses:**

1. Although the problem is interesting, the proposed methodology is not surprising. The main idea is to use PPR to calculate probability and select the top of entities and relations. The reasoning on the subgraph is following the existing methods.
2. The paper emphasizes improved efficiency, but I did not find a comparison of efficiency between the proposed method and the existing ones.
3. The hyperparameter searching looks inefficient.

**Questions:**

1. The hyperparameter searching is not trivial. It would be helpful if the authors could explain more about hyperparameter searching when facing a new knowledge graph and how to select the best configuration.

---

> ### Author Response · Authors · 2023-11-19
> **Response to Reviewer RRjT (1/3)**
>
> We thank reviewer RRjT for the valuable feedback. We addressed all the comments you provided. Please find the point-to-point responses below. Any further comments and discussions are welcome!
>
> **Q1.** About the novelty and contribution.
>
> > Although the problem is interesting, the proposed methodology is not surprising. The main idea is to use PPR to calculate probability and select the top of entities and relations. The reasoning on the subgraph is following the existing methods.
>
> **Reply:** Thanks for the feedback. We would like to kindly argue that it is **non-trivial** to achieve efficient and adaptive link prediction on large-scale KGs like YAGO3-10 and OGBL-WIKIKG2. Existing methods can easily get **out of memory** on these benchmarks, and searching for adaptive configuration is also **infeasible**. The proposed method in this work is **general** to existing methods for KG link prediction and can be integrated with more heuristic indicators for sampling.
>
> Here, we would further explain the novelty and contributions in the following three folds:
>
> - **A valuable research problem.** Our investigation delves into the limitations of structural models, which confront a pronounced scalability challenge. These models rely on prediction over the entire knowledge graph, encompassing all entities and edges, with the additional burden of scoring all entities as potential answers. This approach proves to be highly inefficient, impeding their optimization when applied to large-scale KGs. As a consequence, we present an open question, pondering how to conduct prediction on knowledge graphs efficiently and effectively, and seeking avenues to overcome this hindrance.
>
> - **A conceptual framework with several practical instantiations.** In response to the prevalent scalability challenges faced by existing KG methods, we present a conceptual solution, i.e., the one-shot-subgraph manner for link prediction. This approach owns the enhanced flexibility and efficacy in design. Specifically, instead of conducting direct prediction on the complete original KG, we advocate a two-step prediction process involving subgraph sampling and subgraph-based prediction. In this regard, our proposed prediction manner comprises a sampler and a predictor. Leveraging the efficiency gains derived from subgraph-based prediction, we introduce an optimization technique for subgraph-based searching, incorporating several well-crafted technical designs. These innovations aim to strike a balance between prediction efficiency and the complexity associated with identifying optimal configurations within both data and model spaces.
>
> - **Several important discoveries from experiments.** Through comprehensive experimentation on five prevalent KGs, we demonstrate that our framework achieves state-of-the-art performance. Particularly noteworthy are the substantial advancements we achieve in both efficiency and effectiveness, a trend that is particularly evident in the case of the largest YAGO3-10 and OGB datasets. Our quest for optimal configurations leads us to the intriguing revelation that utilizing the entire KG for prediction is unnecessary. Instead, simple heuristics can efficiently identify a small proportion of entities and facts essential for answering specific queries without the need for additional learning. These compelling findings hold significant meaning and are poised to pique the interest of the KG community.

---

> ### Author Response · Authors · 2023-11-19
> **Response to Reviewer RRjT (2/3)**
>
> **Q2.** The comparison of efficiency.
>
> > The paper emphasizes improved efficiency, but I did not find a comparison of efficiency between the proposed method and the existing ones.
>
> **Reply:** Thanks for the comment. As can be seen in Tab. 6 of our original draft, we provide a detailed efficiency comparison between our one-shot-subgraph method (with 10% entities) and two SOTA methods, i.e., NBFNet and RED-GNN (the vanilla implementation with 100% entities). **Tab. 6 below shows that the training time is significantly reduced when learning with our method.** Notably, on the large dataset YAGO3-10, **94.3%** and **94.5%** of training time (for one epoch) can be saved for NBFNet and RED-GNN, respectively.
>
> | Table 6                          | WN18RR    |          |          |               | YAGO3-10  |          |          |               |
> | ----------------------------------------- | --------- | -------- | -------- | ------------- | --------- | -------- | -------- | ------------- |
> | Metrics                                   | MRR       | H@1      | H@10     | Training Time | MRR       | H@1      | H@10     | Training Time |
> | NBFNet (100% entities)                    | 0.551     | 49.7     | **66.6** | 32.3 min      | 0.550     | 47.9     | 68.3     | 493.8 min     |
> | NBFNet + one-shot-subgraph (10% entities) | **0.554** | **50.5** | 66.3     | **2.6 min**   | **0.565** | **49.6** | **69.2** | **28.2 min**  |
> | RED-GNN (100% entities)                   | 0.533     | 48.5     | 62.4     | 68.7 min      | 0.559     | 48.3     | 68.9     | 1382.9 min    |
> | RED-GNN+ one-shot-subgraph (10% entities) | **0.567** | **51.4** | **66.6** | **4.5 min**   | **0.606** | **54.0** | **72.1** | **76.3 min**  |
>
> Besides, we conduct an efficiency study to investigate the improvement of efficiency brought by the proposed method. The training time and GPU memory of an $8$-layer GNN are summarized in Tab. 7 (see below). As can be seen, a notable advantage of our method is that it has a lower computing cost in terms of running time and memory cost. Particularly on the YAGO3-10 dataset, the existing GNN-based methods will run out of memory (OOM) with a deep architecture. **However, with our one-shot-subgraph sampling of lower ratios of $r^q_{\mathcal{V}}$ and $r^q_{\mathcal{E}}$, the learning and prediction of GNNs become feasible with less memory cost and achieving state-of-the-art performance.**
>
> | Table 7   |                     |                     | WN18RR |        | NELL-995 |        | YAGO3-10 |        |
> | --------- | ------------------- | ------------------- | ------ | ------ | -------- | ------ | -------- | ------ |
> | phase     | $r^q_{\mathcal{V}}$ | $r^q_{\mathcal{E}}$ | Time   | Memory | Time     | Memory | Time     | Memory |
> | Training  | 1.0                 | 1.0                 | OOM    | OOM    | OOM      | OOM    | OOM      | OOM    |
> | Training  | 0.5                 | 0.5                 | 26.3m  | 20.3GB | 1.6h     | 20.1GB | OOM      | OOM    |
> | Training  | 0.2                 | 1.0                 | 12.8m  | 20.2GB | 1.2h     | 18.5GB | OOM      | OOM    |
> | Training  | 0.2                 | 0.2                 | 6.7m   | 6.4GB  | 0.6h     | 8.9GB  | 2.1h     | 23.1GB |
> | Training  | 0.1                 | 1.0                 | 7.2m   | 9.8GB  | 0.8h     | 12.1GB | 1.3h     | 13.9GB |
> | Training  | 0.1                 | 0.1                 | 6.6m   | 5.1GB  | 0.3h     | 5.3GB  | 0.9h     | 10.2GB |
> | Inference | 1.0                 | 1.0                 | 7.3m   | 6.7GB  | 17.5m    | 12.8GB | 1.6h     | 15.0GB |
> | Inference | 0.5                 | 0.5                 | 6.0m   | 4.3GB  | 8.3m     | 4.5GB  | 1.1h     | 10.1GB |
> | Inference | 0.2                 | 1.0                 | 3.2m   | 5.8GB  | 4.2m     | 12.1GB | 0.7h     | 14.7GB |
> | Inference | 0.2                 | 0.2                 | 2.8m   | 1.9GB  | 3.6m     | 2.5GB  | 0.6h     | 3.7GB  |
> | Inference | 0.1                 | 1.0                 | 2.7m   | 2.7GB  | 3.1m     | 9.4GB  | 0.4h     | 9.7GB  |
> | Inference | 0.1                 | 0.1                 | 2.3m   | 1.7GB  | 2.9m     | 1.9GB  | 0.4h     | 3.1GB  |

---

> ### Author Response · Authors · 2023-11-19
> **Response to Reviewer RRjT (3/3)**
>
> **Q3.** About the hyperparameter searching.
>
> > The hyperparameter searching looks inefficient. The hyperparameter searching is not trivial. It would be helpful if the authors could explain more about hyperparameter searching when facing a new knowledge graph and how to select the best configuration.
>
> **Reply:** Thanks for the valuable feedback. We would like to explain the detailed hyper-parameter searching process from three aspects, i.e., search space, search problem, and search algorithm. For a better understanding, we also add Fig. 2 to illustrate the technical details.
>
> **The search space.** We denote the sampler's hyper-parameters as $\phi_{\text{hyper}}$ and the predictor's hyper-parameters as $\theta_{\text{hyper}}$. Specifically, $\phi_{\text{hyper}}$ contains $(r^q_{\mathcal{V}}, r^q_{\mathcal{E}})$, while $\theta_{\text{hyper}}$ contains several intra-layer or inter-layer designs, as illustrated in Fig. 2(a).
>
> **The search problem**. Next, we propose the bi-level optimization problem to adaptively search for the optimal configuration $(\phi^*_{\text{hyper}}, \theta^*_{\text{hyper}})$ of design choices on a specific KG, namely,
> $$
> \phi^*_{\text{hyper}} = \arg \max_{ \phi_{\text{hyper}}} \mathcal{M}( f_{(\theta_{\text{hyper}}^*, \theta_{\text{learn}}^*)}, g_{ \phi_{\text{hyper}}}, \mathcal{E}^{\text{val}}), \\
> 				 \text{s.t. }
> 				 \theta_{\text{hyper}}^*  =
> 				\arg \max_{ \theta_{\text{hyper}}}
> 				\mathcal{M}(f_{( \theta_{\text{hyper}},  \theta_{\text{learn}}^*)}, g_{\bar{ \phi}_{\text{hyper}}}, \mathcal{E}^{\text{val}}),
> $$
>
> where the performance measurement $\mathcal{M}$ can be Mean Reciprocal Ranking (MRR) or Hits@k. Note the non-parametric sampler $g_\phi$ only contains hyper-parameters $\phi_{\text{hyper}}$. As for predictor $f_\theta$, its $\theta   =   (\theta_{\text{hyper}}, \theta_{\text{learn}})$ also includes the learnable parameter $\theta_{\text{learn}}$ that $\theta_\text{learn}^*   =   \arg \min_{\theta_{\text{learn}}} \mathcal{L} (f_{(\theta_\text{hyper}, \theta_\text{learn})}, g_{\bar\phi_{\text{hyper}}}, \mathcal{E}^{\text{train}})$.
>
> **The search algorithm.** Directly searching on both data and model spaces is expensive due to the large space size and data scale. Hence, we split the search into two sub-processes as Fig. 2(b),
>
> - First, we freeze the sampler $g_{\bar\phi}$ (with constant $\phi_{\text{hyper}}$) to search for the optimal predictor $f_{\theta^*}$ with (1) the hyper-parameters optimization for $\theta^*_{\text{hyper}}$ and (2) the stochastic gradient descent for $\theta^*_{\text{learn}}$.
>
> - Then, we freeze the predictor $f_{\theta^*}$ and search for the optimal sampler $g_{\phi^*}$, simplifying to pure hyper-parameters optimization for $\phi^*_{\text{hyper}}$ in a zero-gradient manner with low computation complexity.
>
> Specifically, we follow the sequential model-based Bayesian Optimization (BO) to obtain $\phi^*_{\text{hyper}}$ and $\theta^*_{\text{hyper}}$. Random forest (RF) is chosen as the surrogate model because it has a stronger power for approximating the complex and discrete curvature compared with other common surrogates, e.g., Gaussian Process (GP) or Multi-layer Perceptron (MLP).
>
> Besides, we also devise a data split trick that balances observations and predictions. It saves time as the training does not necessarily traverse all the training facts. Specifically, partial training facts are sampled as training queries while the others are treated as observations, i.e., we randomly separate the training facts into two parts as $\mathcal{E}^{\text{train}}   =   \mathcal{E}^{\text{obs}} \cup \mathcal{E}^{\text{query}}$, where the overall prediction system $f_\theta \circ g_\phi$ takes $\mathcal{E}^{\text{obs}}$ as input and then predicts $\mathcal{E}^{\text{query}}$ (see Fig. 2(c)). Here, the split ratio $r^{\text{split}}$ is to balance the sizes of these two parts as $r^{\text{split}}   =   \frac{|\mathcal{E}^{\text{obs}}|}{|\mathcal{E}^{\text{query}}|}$. Thus, the training becomes $\theta^*   =  \arg\min_\theta \Sigma_{(u, q, v) \in \mathcal{E}^{\text{query}}} \mathcal{L} \big( f_\theta(G_s), v \big)$  with the split query edges $\mathcal{E}^{\text{query}}$, where $G_s = g_\phi (\mathcal{E}^{\text{obs}} , u, q)$ with the split observation edges $\mathcal{E}^{\text{obs}}$.
>
> The above contents have been added to Sec 4.2 of our latest draft and highlighted in blue.

---

> ### Author Response · Authors · 2023-11-22
> **We are anticipating your post-rebuttal feedback!**
>
> Dear Reviewer RRjT,
>
> Thanks very much for your time and valuable comments.
>
> In the rebuttal period, we have provided detailed responses to all your comments and questions and made significant modifications in the revision.
>
> **We understand you might be quite busy. However, as the discussion deadline is approaching, would you mind checking our response and confirming whether you have any further questions?**
>
> Any comments and discussions are welcome!
>
> Thanks for your attention and best regards.
>
> Authors of #3620

---

> ### Author Response · Authors · 2023-11-23
> **[Last-day Reminder] We are anticipating your post-rebuttal feedback!**
>
> Dear Reviewer RRjT,
>
> Thanks very much for your time and valuable comments.
>
> We understand you might be quite busy. **However, the discussion deadline is approaching, and we have only a few hours left.**
>
> Would you mind checking our response and confirming whether you have any further questions?
>
> Thanks for your attention.
>
> Best regards,
>
> Authors of #3620

---

### Official Review · Reviewer_7ny9 · 2023-11-10

**Soundness:** 2 fair
**Presentation:** 2 fair
**Contribution:** 2 fair
**Rating:** 3
**Confidence:** 4

**Summary:**

This paper introduces a new method for knowledge graph (KG) link prediction which they name reasoning. The method relies on (1) sampling the KG around the entities involved (not on e.g. computing embedding on the whole KG) and (2) reasoning on top of this subgraph. This two step process ensures efficiency as the subgraph reduces the neighborhood over which the reasoning (link prediction) is done. Step (1) involved a Page Rank based procedure that selected a subgraph based on the importance of the nodes in the KG.

**Strengths:**

* The paper proposes a simple, yet somewhat novel idea. While the idea is somewhat heuristic, it is nevertheless interesting to test out its performance on this problem.

* The authors perform good literature review summarizing the different categories of KG link prediction methods.

* The proposal is tested empirically through several experiments.

**Weaknesses:**

* The paper appears to use terms and phrases that make it sound quite bombastic and somewhat unscientific (see below).

* Some of the language is vague and I feel like some of the technical details from the appendix would benefit the paper.

* Algorithm 1 is not clearly defined (see below).

* I feel like a bit more space can be allocated to the intuition behind some of the experiments (e.g. what does the degree distribution in Fig 4 imply about the method's performance qualitatively?)

**Questions:**

* To the above point, re language usage, please fix these.
    * E.g. it uses the term KG Reasoning while in fact solving the link prediction task (a term used only once in the whole paper!).
    * Repeated use of the word paradigm, even though there isn't anything that is fundamentally new theory in their proposal/approach.
    * The use of the word 'blazes' on p.1

* p. 3, Advantage 2. Why is the exploration/exploitation relevant for the proposed algorithms? While there is some intution behind the analogy with exploitaion/exploration the algorithm doesn't really formulate the problem using this analogy. For example there is no way to trade/balance the two. So this comparison relies on broad intuition, hence this advantage is a weak one at best.

* Algorithm 1. define Alpha, K, define the shape of A and D.

* How is the adjacency matrix A computed in the subgraph extraction part. Since this is a KG, not a homogenous graph, there are several adjacency matrices (one for each relation). Are these combined somehow?

* I think the paper would benefit from pulling some of the technical details from the supplement to the main paper, in favor of reducing some of the verbiage in the main paper. Please consider this modification.

---

> ### Author Response · Authors · 2023-11-19
> **Response to Reviewer 7ny9 (1/2)**
>
> We thank reviewer 7ny9 for the valuable feedback. We addressed all the comments you provided. Please find the point-to-point responses below. Any further comments and discussions are welcome!
>
> **Q1.** About the writing and novelty.
>
> > Q1.1 The paper appears to use terms and phrases that make it sound quite bombastic and somewhat unscientific.
> > Q1.2 Some of the language is vague and I feel like some of the technical details from the appendix would benefit the paper.
> > Q1.3 It uses the term KG Reasoning while in fact solving the link prediction task (a term used only once in the whole paper!).
> > Q1.4 Repeated use of the word paradigm, even though there isn't anything that is fundamentally new theory in their proposal/approach.
> > Q1.5 The use of the word 'blazes' on p.1.
> > Q1.6 I think the paper would benefit from pulling some of the technical details from the supplement to the main paper, in favor of reducing some of the verbiage in the main paper. Please consider this modification.
> > Q1.7 I feel like a bit more space can be allocated to the intuition behind some of the experiments (e.g. what does the degree distribution in Fig 4 imply about the method’s performance qualitatively?)
>
> **Reply:** Thanks for your kind suggestions. **We have made several modifications to our draft. Please refer to our latest draft, where the modified contents are highlighted in blue.** Specifically,
>
> - **Word use.** We update the words used in our draft, e.g., "paradigm" -> "manner", "KG reasoning" -> "link prediction", and "reasoning" -> "prediction". The keyword of our method is fine-tuned to "one-shot subgraph link prediction" and formally defined in Def. 1. We delete those repeated claims, e.g., the two-step prediction pipeline with sampling and predicting.
> - **Technical details.** We move some technical details from the appendix to the main contents. Please refer to Sec. 4.2, where the implementation details of search space, search problem, and search algorithm are elaborated. Besides, for a better understanding, we also add Fig. 2 to illustrate the technical details.
> - **Experimental intuitions.** The intuitions of the experiments are further explained and highlighted in Sec. 5. For example, in Fig. 5, the below distributions of degree and distance show the statistical properties of each subgraph. More importantly, the visualization of Fig.5 shows that the local structure of query entity $u$ is well preserved, while the true answers $v$ are also covered in the subgraphs, which contributes to the model's performance.
> - **Mathematical symbols and equations.** We add more description to the used variables in Sec. 4.3 and leave the detailed proof in Appendix A.3. We also label the formulas in Secs. 4.2 and 4.3 with numbers.
>
> We thoroughly consider the questions the reviewer mentioned and try our best to address the concerns. Please tell us if some content needs to be further modified, and we will do it as soon as possible.
>
> Besides, as you also point out the weakness of novelty or originality, we would like to further explain the novelty and contributions in the following three folds:
>
> - **A valuable research problem.** Our investigation delves into the limitations of structural models, which confront a pronounced scalability challenge. These models rely on prediction over the entire knowledge graph, encompassing all entities and edges, with the additional burden of scoring all entities as potential answers. This approach proves to be highly inefficient, impeding their optimization when applied to large-scale KGs. As a consequence, we present an open question, pondering how to conduct prediction on knowledge graphs efficiently and effectively, and seeking avenues to overcome this hindrance.
>
> - **A conceptual framework with several practical instantiations.** In response to the prevalent scalability challenges faced by existing KG methods, we present a conceptual solution, i.e., the one-shot-subgraph manner for link prediction. This approach owns the enhanced flexibility and efficacy in design. Specifically, instead of conducting direct prediction on the complete original KG, we advocate a two-step prediction process involving subgraph sampling and subgraph-based prediction. In this regard, our proposed prediction manner comprises a sampler and a predictor. Leveraging the efficiency gains derived from subgraph-based prediction, we introduce an optimization technique for subgraph-based searching, incorporating several well-crafted technical designs. These innovations aim to strike a balance between prediction efficiency and the complexity associated with identifying optimal configurations within both data and model spaces.

---

> ### Author Response · Authors · 2023-11-19
> **Response to Reviewer 7ny9 (2/2)**
>
> - **Several important discoveries from experiments.** Through comprehensive experimentation on five prevalent KGs, we demonstrate that our framework achieves state-of-the-art performance. Particularly noteworthy are the substantial advancements we achieve in both efficiency and effectiveness, a trend that is evident in the case of the largest YAGO3-10 and OGB datasets. Our quest for optimal configurations leads to the intriguing revelation that utilizing the entire KG for prediction is unnecessary. Instead, simple heuristics can efficiently identify a small proportion of entities and facts essential for answering specific queries without the need for additional learning. These compelling findings hold significant meaning and are poised to pique the interest of the KG community.
>
> **Q2.** Algorithm 1 is not clearly defined.
>
> > Algorithm 1. define Alpha, K, define the shape of A and D.
>
> **Reply:** Thanks for the feedback. We provide a detailed explanation as follows, along with the corresponding modifications in the revised submission.
>
> - The two-dimensional degree matrix is denoted as $D  \in  \mathbb{R}^{|\mathcal{V}| \times |\mathcal{V}|}$ , and adjacency matrix is $A  \in  \{0,1\}^{|\mathcal{V}| \times |\mathcal{V}|}$. These two matrixes together work as the transition matrix in PPR, wherein $A_{ij}  =  1$ means an edge $(i, r, j)  \in  \mathcal{E}$ and $D_{ij}  =  \text{degree}(v_i)$ if $i  =  j$ else $D_{ij}  =  0$. Besides, $K$ is the maximal PPR iterations, and the damping coefficient $\alpha$ ($=0.85$ by default) controls the differentiation degree.
> - The definitions of Alpha, K, A, and D in Algorithm 1 are consistent with Eqn. 3 in Step-1 of Sec. 4.1. We have revised their descriptions in Sec. 4.1 to be clearer and updated the "Require part" of Algorithm 1, including the definitions of all these variables. In addition, Algorithm 1 is moved to the end of introducing three steps. We think the logical flow can be more smooth, and this algorithm provides an overview of the link prediction system.
>
> **Q3.** About the Advantage 2.
>
> > Why is the exploration/exploitation relevant for the proposed algorithms? While there is some intution behind the analogy with exploitaion/exploration the algorithm doesn't really formulate the problem using this analogy. For example there is no way to trade/balance the two. So this comparison relies on broad intuition, hence this advantage is a weak one at best.
>
> **Reply:** Thanks for the insightful feedback. Conceptually, in our case, exploration indicates the range of message propagation w.r.t. the query entity $u$, while exploitation indicates the depth. Structural methods that adopt the massage passing with a BFS manner can only explore and exploit the full L-hop neighbors of u, whose range and depth are exactly the same. By contrast, our paradigm is more flexible in scopes that are not restricted to L-hop, and the range and depth can be different, which is a clear advantage over the common structural models without sampling.
>
> We have simplified and clarified this part to avoid misunderstanding, and the updated Advantage 2 is
>
> > **Advantage 2** (Flexible propagation scope). The scope here refers to the range of message propagation starting from the query entity $u$. Normally, an $L$-layer structural method will propagate to the full $L$-hop neighbors of $u$. By contrast, our one-shot sampling enables the bound of propagation scope within the extracted subgraph, where the scope is decoupled from the model's depth $L$.
>
> Empirically, we found that
>
> - involving the whole entities degenerates the prediction as too many irrelevant entities are covered (see Tab.4)
> - a deeper predictor with a larger $L$ consistently brings better results.
>
> These observations enlighten us to learn a deeper predictor with small subgraphs.
>
> **Q4.** The computation of adjacency matrix.
>
> > How is the adjacency matrix A computed in the subgraph extraction part. Since this is a KG, not a homogenous graph, there are several adjacency matrices (one for each relation). Are these combined somehow?
>
> **Reply:** Thanks for the feedback. Yes, it is exactly your understanding that the adjacencies of different relations are combined. As we introduced in Section 4.1, $A_{ij}  =  1$ means an edge $(i, r, j)  \in  \mathcal{E}$, we combine all the facts of different relations to this two-dimensional adjacent matrix. We have revised this description to be clearer.

---

> ### Author Response · Authors · 2023-11-21
> **Would you mind checking our responses and confirming whether you have any further questions?**
>
> Dear Reviewer 7ny9,
>
> Thanks very much for your time and valuable comments.
>
> In the rebuttal period, we have provided detailed responses to all your comments and questions point-by-point for the unclear presentations. Specifically, we
> - improve the writing to be more clear and informative (**Q1**);
> - clarify the definitions of variables in Algorithm 1 to be clearer (**Q2**);
> - revise the Advantage 2 to be more precise (**Q3**);
> - further explain the computation of the adjacency matrix in PPR (**Q4**).
>
> Would you mind checking our responses and confirming whether you have any further questions?
>
> Any comments and discussions are welcome!
>
> Thanks for your attention and best regards.
>
> Authors of #3620

---

> ### Author Response · Authors · 2023-11-22
> **We are anticipating your post-rebuttal feedback!**
>
> Dear Reviewer 7ny9,
>
> Thanks very much for your time and valuable comments.
>
> In the rebuttal period, we have provided detailed responses to all your comments and questions and made significant modifications in the revision.
>
> **We understand you might be quite busy. However, as the discussion deadline is approaching, would you mind checking our response and confirming whether you have any further questions?**
>
> Any comments and discussions are welcome!
>
> Thanks for your attention and best regards.
>
> Authors of #3620

---

> ### Author Response · Authors · 2023-11-23
> **[Last-day Reminder] We are anticipating your post-rebuttal feedback!**
>
> Dear Reviewer 7ny9,
>
> Thanks very much for your time and valuable comments.
>
> We understand you might be quite busy. **However, the discussion deadline is approaching, and we have only a few hours left.**
>
> Would you mind checking our response and confirming whether you have any further questions?
>
> Thanks for your attention.
>
> Best regards,
>
> Authors of #3620

---

> > ### Comment · Reviewer_7ny9 · 2023-12-04
> > **comments on the authors' response**
> >
> > Dear authors, thank you very much for all the comments and the work put towards all the modifications. Your comments address my questions and I am willing to raise my score by one point. That said, I will keep an eye on the other reviewers responses as I think they are brining in very valid points and concerns.

---

### Author Response · Authors · 2023-11-19
**A General Response by Authors**

**We would like to thank all the reviewers for their thoughtful suggestions on our paper.**

**We are glad that the reviewers generally have positive impressions of our work**, including **(1)** a novel and reasonable idea (7ny9, RRjT, FPeS), **(2)** a comprehensive literature review (7ny9, RRjT), **(3)** thorough experiments and promising results (7ny9, FPeS), and **(4)** well-written and good presentation (RRjT, FPeS).

**In the rebuttal period, we have provided detailed responses to all the comments and questions point-by-point.** Specifically, we further improve the writing to be more clear and informative (Q1 for 7ny9, Q1 for FPeS), clarify the key contributions (Q1 for 7ny9, Q1 for RRjT), explain the technical details (Q2, Q3, Q4 for 7ny9, Q3 for RRjT), provide empirical results on efficiency comparison (Q2 for RRjT) and data perturbations (Q3 for FPeS), and discuss the intrinsic interpretability (Q2 for FPeS) as well as potential directions (Q4 for FPeS).

Lastly, we would appreciate all reviewers’ time again. Would you mind checking our response and confirming whether you have any further questions? **We are anticipating your post-rebuttal feedback!**

---

### Meta-Review · Area_Chair_NZwj · 2023-12-21

**Metareview:**

This paper studies link prediction on large-scale knowledge graphs and recognizes that existing methods utilize the whole KG for prediction and suffer from a severe scalability problem. To address the problem, it proposes a two-step procedure where the first step extracts only one query-dependent subgraph while the second step predicts on the single subgraph. The reviewers generally agree that the proposed method is reasonable and shows promising results on several benchmark datasets, the paper provides a comprehensive literature review. On the other hand, some of them also point out a lot of writing issues, lack of discussions on the insights of PPR sampling, and hyperparameter searching. It seems the authors have spent a lot of efforts revising the paper and addressing the comments with new results and detailed explanations. That said, I think the research work should have been presented and conducted very well at the submission time with probably only minor revisions needed at the rebuttal time. I would recommend accepting the paper but would not mind if it gets rejected due to other papers with higher novelty and more contributions.

**Justification For Why Not Higher Score:**

Please see the weaknesses summarized above.

**Justification For Why Not Lower Score:**

Please see the strengths summarized above.

---

### Decision · Program_Chairs · 2024-01-16

Accept (poster)